# Predicting the sequence-dependent backbone dynamics of intrinsically disordered proteins

**Sanbo Qin[1], Huan-Xiang Zhou[1,2]\***

[1]Department of Chemistry, University of Illinois Chicago, Chicago, United States; [2]Department of Physics, University of Illinois Chicago, Chicago, United States

## eLife assessment

In this **useful** study, a **solid** machine learning approach based on a broad set of systems to predict the R2 relaxation rates of residues in intrinsically disordered proteins (IDPs) is described. The ability to predict the patterns of R2 will be helpful to guide experimental studies of IDPs. A potential weakness is that the predicted R2 values may include both fast and slow motions, thus the predictions provide only limited new physical insights into the nature of the underlying protein dynamics, such as the most relevant timescale.

**\*For correspondence:**
hzhou43@uic.edu

**Competing interest:** The authors declare that no competing interests exist.

**Abstract** How the sequences of intrinsically disordered proteins (IDPs) code for functions is still an enigma. Dynamics, in particular residue-specific dynamics, holds crucial clues. Enormous efforts have been spent to characterize residue-specific dynamics of IDPs, mainly through NMR spin relaxation experiments. Here, we present a sequence-based method, SeqDYN, for predicting residue-specific backbone dynamics of IDPs. SeqDYN employs a mathematical model with 21 parameters: one is a correlation length and 20 are the contributions of the amino acids to slow dynamics. Training on a set of 45 IDPs reveals aromatic, Arg, and long-branched aliphatic amino acids as the most active in slow dynamics whereas Gly and short polar amino acids as the least active. SeqDYN predictions not only provide an accurate and insightful characterization of sequence-dependent IDP dynamics but may also serve as indicators in a host of biophysical processes, including the propensities of IDP sequences to undergo phase separation.

## Introduction

Intrinsically disordered proteins (IDPs) or regions (IDRs) do not have the luxury of a three-dimensional structure to help decipher the relationship between sequence and function. Instead, dynamics has emerged as a crucial link between sequence and function for IDPs (*Dey et al., 2022*). Nuclear magnetic resonance (NMR) spin relaxation is a uniquely powerful technique for characterizing IDP dynamics, capable of yielding residue-specific information (*Camacho-Zarco et al., 2022*). Backbone [15]N relaxation experiments typically yield three parameters per residue: transverse relaxation rate ($R_2$), longitudinal relaxation rate ($R_1$), and steady-state heteronuclear Overhauser enhancement (NOE). While all three parameters depend on ps-ns dynamics, $R_2$ is the one most affected by slower dynamics (10 s of ns to 1 μs). An increase in either the timescale or the amplitude of slower dynamics results in higher $R_2$. For IDPs, $R_2$ is also the parameter that exhibits the strongest dependence on sequence (*Dey et al., 2022*; *Camacho-Zarco et al., 2022*).

$R_2$ was noted early on as an important indicator of residual structure in the unfolded state of the structured protein lysozyme (*Klein-Seetharaman et al., 2002*). This property has since been measured

for many IDPs to provide insight into various biophysical processes. Just as the residual structure in the unfolded state biases the folding pathway of lysozyme (*Klein-Seetharaman et al., 2002*), a nascent α-helix in the free state of Sendai virus nucleoprotein C-terminal domain (Sev-NT), as indicated by highly elevated $R_2$ (*Abyzov et al., 2016*), biases the coupled binding and folding pathway in the presence of its target phosphoprotein (*Schneider et al., 2015*). Local secondary structure preformation also facilitates the binding of yes-associated protein (YAP) with its target transcription factor (*Feichtinger et al., 2022*). Likewise a correlation has been found between $R_2$ in the free state and the membrane binding propensity of synaptobrevin-2: residues with elevated $R_2$ have increased propensity for membrane binding (*Lakomek et al., 2019*). $R_2$ in the free state has also been used to uncover factors that promote liquid-liquid phase separation of IDPs. For example, a nascent α-helix (shown by elevated $R_2$) is important for the phase separation of the TDP-43 low-complexity domain, as both the deletion of the helical region and a helix-breaking mutation (Ala to Pro) abrogates phase separation (*Conicella et al., 2016*). Similarly, nascent α-helices in the free state of cytosolic abundant heat-soluble 8 (CAHS-8), upon raising concentration and lowering temperature stabilize to form the core of fibrous gels (*Malki et al., 2022*). For the hnRNPA1 low-complexity domain (A1-LCD), aromatic residues giving rise to local peaks in $R_2$ also mediate phase separation (*Martin et al., 2020*).

Both NMR relaxation data and molecular dynamics (MD) simulations have revealed determinants of $R_2$ for IDPs. It has been noted that the flexible Gly tends to lower $R_2$, whereas secondary structure and contact formation tend to raise $R_2$ (*Cook et al., 2019*). This conclusion agrees well with recent MD simulations (*Dey et al., 2022*; *Hicks et al., 2020*; *Yu and Brüschweiler, 2022*; *Smrt et al., 2023*). These MD studies, using IDP-specific force fields, are able to predict $R_2$ in quantitative agreement with NMR measurements, without ad hoc reweighting as done in earlier studies. According to MD, most contact clusters are formed by local sequences, within blocks of up to a dozen or so residues (*Dey et al., 2022*; *Hicks et al., 2020*; *Smrt et al., 2023*). Tertiary contacts can also form but are relatively rare; as such their accurate capture requires extremely extensive sampling and still poses a challenge for MD simulations. Contrary to Gly, aromatic residues have been noted as mediators of contact clusters (*Klein-Seetharaman et al., 2002*; *Martin et al., 2020*).

*Schwalbe et al., 1997* introduced a mathematical model to describe the $R_2$ profile along the sequence for lysozyme in the unfolded state. The $R_2$ value of a given residue was expressed as the sum of contributions from this residue and its neighbors. This model yields a mostly flat profile across the sequence, except for a falloff at the termini, resulting in an overall bell shape. *Klein-Seetharaman et al., 2002* then fit peaks above this flat profile as a sum of Gaussians. *Cho et al., 2007* proposed bulkiness as a qualitative indicator of backbone dynamics. Recently *Sekiyama et al., 2022* calculated $R_2$ as the geometric mean of 'indices of local dynamics'; the latter were parameterized by fitting to the measured $R_2$ for a single IDP. All these models merely describe the $R_2$ profile of a given IDP, and none of them is predictive.

Here, we present a method, SeqDYN, for predicting $R_2$ of IDPs. Using a mathematical model introduced by *Li et al., 2020* to predict propensities for binding nanoparticles and also adapted for predicting propensities for binding membranes (*Qin et al., 2022*), we express the $R_2$ value of a residue as the product of contributing factors from all residues. The contributing factor attenuates as the neighboring residue becomes more distant from the central residue. The model, after training on a set of 45 IDPs, has prediction accuracy that is competitive against that of the recent MD simulations using IDP-specific force fields (*Dey et al., 2022*; *Hicks et al., 2020*; *Yu and Brüschweiler, 2022*; *Smrt et al., 2023*). For lysozyme and other structured proteins, the SeqDYN prediction agrees remarkably well with $R_2$ measured in their unfolded state.

## Results
### The data set of IDPs with $R_2$ rates
We collected $R_2$ data for a total of 54 nonhomologous IDPs or IDRs (*Table 1*; *Figure 1*). According to indicators from NMR properties, including low or negative NOEs, narrow dispersion in backbone amide proton chemical shifts, and small secondary chemical shifts (SCSs), most of the proteins are disordered with at most transient α-helices. A few are partially folded, including Sev-NT with a well-populated (~80%) long helix (residues 478–491; *Jensen et al., 2008*), CREB-binding protein fourth intrinsically disordered linker (CBP-ID4) with >50% propensities for two long helices (residues 2–25

**Table 1.** Experimental conditions, mean and standard deviation of measured $R_2$, and SeqDYN prediction RMSE.

| Protein name | # of res | Temp (K) | $B_0$ (MHz) | $\bar{R}_2$ (s$^{-1}$) | $\sigma_{R_2}$ (s$^{-1}$) | RMSE (s$^{-1}$) | PMID; ref |
|---|---|---|---|---|---|---|---|
| Training set (45 IDPs) [*] | | | | | | | |
| A1-LCD | 131 | 298 | 800 | 2.68 | 0.46 | 0.60 | 32029630; *Martin et al., 2020* |
| Aβ40 | 40 | 278 | 600 | 3.40 | 0.92 | 0.38 | 31181936; *Rezaei-Ghaleh et al., 2019* |
| Ash1 | 83 | 278 | 800 | 9.80 | 1.40 | 1.41 | 27807972; *Martin et al., 2016* |
| Beclin1 | 165 | 288 | 800 | 5.37 | 1.03 | 1.14 | 27288992; *Yao et al., 2016* |
| CAPRIN1 | 103 | 303 | 600 | 5.34 | 0.88 | 0.72 | 31898464; *Wong et al., 2020* |
| CBP-ID4 | 207 | 283 | 700 | 5.45 | 2.55 | 2.01;1.90[†] | 29790640; *Murrali et al., 2018* |
| GbnD4-DHD | 91 | 280 | 700 | 6.81 | 1.55 | 1.28 | 29309054; *Jenner et al., 2018* |
| ERD14 | 185 | 288 | 600 | 3.96 | 0.87 | 0.54 | 21336827; *Szalainé Ágoston et al., 2011* |
| ExsE | 88 | 298 | 600 | 3.18 | 0.88 | 0.76 | 22138394; *Zheng et al., 2012* |
| FCP1 | 85 | 298 | 500 | 2.94 | 0.54 | 0.43 | 26286791; *Lawrence and Showalter, 2012* |
| FUS | 163 | 298 | 850 | 3.48 | 0.51 | 0.54 | 26455390; *Burke et al., 2015* |
| GAb1 | 82 | 298 | 500 | 3.99 | 0.88 | 0.89 | 34929201; *Gruber et al., 2022* |
| hACTR | 69 | 304 | 600 | 3.26 | 0.47 | 0.49 | 18177052; *Ebert et al., 2008* |
| Hahellin | 92 | 298 | 800 | 9.94 | 2.69 | 2.85 | 24671380; *Patel et al., 2014* |
| hCSD1 | 141 | 298 | 500 | 3.56 | 0.93 | 0.99 | 18537264; *Kiss et al., 2008* |
| HOX-DFD | 90 | 298 | 600 | 6.98 | 3.15 | 1.99 | 30802457; *Maiti et al., 2019* |
| hZIP4-ICL2 | 100 | 283 | 800 | 9.54 | 2.37 | 1.58 | 30793391; *Bafaro et al., 2019* |
| Jaburetox | 94 | 298 | 800 | 6.01 | 2.30 | 2.27 | 25605001; *Lopes et al., 2015* |
| KRS-NT | 72 | 303 | 600 | 3.26 | 0.93 | 0.83 | 24983501; *Cho et al., 2014* |
| MBP-xα2 | 70 | 295 | 600 | 3.83 | 0.60 | 0.54 | 25343306; *De Avila et al., 2014* |
| MKK4 | 86 | 278 | 850 | 4.49 | 1.42 | 0.63 | 29276882; *Delaforge et al., 2018* |
| N-Cby | 63 | 298 | | 4.19 | 1.20 | 1.25 | 21182262; *Mokhtarzada et al., 2011* |
| Niv-P$_{NTD}$ | 406 | 288 | 700 | 5.41 | 1.82 | 1.66 | 33177626; *Schiavina et al., 2020* |
| NS5A-D2D3 | 268 | 278 | 800 | 8.62 | 3.85 | 2.14 | 26445449; *Sólyom et al., 2015* |
| NUPR1 | 93 | 298 | 600 | 2.98 | 0.82 | 0.76 | 31325636; *Neira et al., 2019* |
| OPN | 220 | 310 | 800 | 2.59 | 0.82 | 0.54 | 31794728; *Mateos et al., 2020* |
| p53TAD | 73 | 298 | 850 | 2.72 | 0.66 | 0.33 | 30240067; *Xie et al., 2018* |
| PDEγ | 87 | 298 | | 3.96 | 1.05 | 0.71 | 18230733; *Song et al., 2008* |
| PKIα | 75 | 300 | 900 | 3.41 | 0.87 | 0.52 | 32338601; *Olivieri et al., 2020* |
| Mev-P$_{NTD}$ | 304 | 298 | 950 | 2.92 | 0.59 | 0.48 | 30140745; *Milles et al., 2018* |
| ProTα | 113 | 283 | 800 | 3.40 | 0.56 | 0.43 | 29466338; *Borgia et al., 2018* |
| Pup | 64 | 298 | 850 | 2.66 | 0.51 | 0.43 | 30240067; *Xie et al., 2018* |
| rmBG21 | 199 | 300 | 600 | 4.06 | 0.90 | 0.63 | 17676872; *Ahmed et al., 2007* |
| RPB1 | 201 | 277 | 850 | 6.48 | 1.74 | 1.33 | 28945358; *Janke et al., 2018* |
| securin | 202 | 283 | 500 | 5.49 | 1.13 | 1.08 | 19053469; *Csizmok et al., 2008* |
| Sev-NT | 124 | 298 | 600 | 3.20 | 1.42 | 0.76;0.38[†] | 27112095; *Abyzov et al., 2016* |
| Sic1 | 92 | 278 | 500 | 3.34 | 0.59 | 0.48 | 20399186; *Mittag et al., 2010* |

*Table 1 continued on next page*

*Table 1 continued*

| Protein name | # of res | Temp (K) | B₀ (MHz) | $\bar{R}_2$ (s⁻¹) | $\sigma_{R_2}$ (s⁻¹) | RMSE (s⁻¹) | PMID; ref |
|---|---|---|---|---|---|---|---|
| SKIPN | 71 | 298 | | 5.64 | 1.05 | 1.46 | 20007319; *Wang et al., 2010* |
| SLBP-NT | 113 | 298 | 600 | 3.96 | 1.40 | 1.61 | 15260482; *Thapar et al., 2004* |
| α-synuclein | 140 | 298 | 600 | 2.96 | 0.53 | 0.44 | 30184304; *Rezaei-Ghaleh et al., 2018* |
| SOCS5-JIR | 70 | 303 | 800 | 4.32 | 2.36 | 1.91 | 26173083; *Chandrashekaran et al., 2015* |
| tau K18 | 129 | 283 | 700 | 4.12 | 0.95 | 0.83 | 23740819; *Barré and Eliezer, 2013* |
| TC1 | 106 | 298 | 600 | 4.65 | 1.61 | 1.24 | 23189168; *Cino et al., 2012* |
| TDP-43 | 151 | 283 | 500 | 4.07 | 1.51 | 0.96 | 27545621; *Conicella et al., 2016* |
| γ-tubulin-CT | 39 | 288 | 500 | 2.23 | 0.35 | 0.27 | 29127738; *Harris et al., 2018* |
| Test set (9 IDPs) | | | | | | | |
| AMOTL1 | 207 | 283 | 800 | 8.45 | 2.55 | 2.04 | 35481651; *Vogel et al., 2022* |
| CAHS-8 | 233 | 303 | 850 | 4.43 | 3.25 | 2.36;1.92[†] | 34750927; *Malki et al., 2022* |
| ChiZ | 64 | 298 | 800 | 4.33 | 0.89 | 0.74 | 32585849; *Hicks et al., 2020* |
| α-endosulfine | 121 | 298 | 800 | 3.21 | 0.81 | 0.48 | 34346186; *Thapa et al., 2022* |
| FtsQ | 99 | 305 | 800 | 6.44 | 3.78 | 2.32;1.71[†] | 36959324; *Smrt et al., 2023* |
| Pdx1 | 83 | 298 | 500 | 2.98 | 0.70 | 0.76 | 30525611; *Cook et al., 2019* |
| synaptobrevin-2 | 96 | 278 | 600 | 5.54 | 1.80 | 0.72 | 30975750; *Lakomek et al., 2019* |
| TIA-1 | 91 | 310 | 800 | 4.01 | 0.89 | 0.55 | 36112647; *Sekiyama et al., 2022* |
| YAP | 122 | 298 | 800 | 3.19 | 1.44 | 1.23 | 35378854; *Feichtinger et al., 2022* |

[*]For training set, RMSE is calculated for prediction based on leave-one-out training (using 44 IDPs).

[†]First number is for SeqDYN prediction; second number is after applying a helix boost.

and 101–128; *Piai et al., 2016*), HOX transcription factor DFD (HOX-DFD) with a well-folded domain comprising three helices (*Maiti et al., 2019*), and Hahellin (apo form) as a molten globule (*Patel et al., 2014*). In *Figure 2*, we display representative conformations of five IDPs, ranging from fully disordered MAPK kinase 4 (MKK4; *Delaforge et al., 2018*) and α-synuclein (*Sung and Eliezer, 2007*) to Measles virus phosphoprotein N-terminal domain (Mev-P_NTD; *Milles et al., 2018*) with transient short helices to Sev-NT and CBP-ID4 with stable long helices. The sequences of all the IDPs are listed in Appendix 1.

We used 45 of the 54 IDPs to train and validate SeqDYN and reserved the remaining 9 for testing. The sequence lengths of the training set range from 39 to 406 residues, with an average of 125.3 residues. Altogether $R_2$ data are available for 3966 residues. A large majority (35 out of 45) of the 45 IDPs have mean $R_2$ values ($\bar{R}_2$, calculated among all the residues in a protein) between 2.5 and 5.5 s⁻¹ (*Table 1* and *Figure 3A*). This $\bar{R}_2$ range is much lower than that of structured proteins with similar sequence lengths. The low $\bar{R}_2$ values and lack of dependence on sequence length (*Figure 3—figure supplement 1A*) suggest that $R_2$ of the IDPs is mostly dictated by local sequence instead of tertiary interaction.

The most often used temperature for acquiring the $R_2$ data was 298 K, but low temperatures (277–280 K) were used in a few cases (*Table 1* and *Figure 3—figure supplement 1B*). Of the seven IDPs with $\bar{R}_2 > 6.4$ s⁻¹, four can be attributed to low temperatures (*Sólyom et al., 2015*; *Martin et al., 2016*; *Janke et al., 2018*; *Jenner et al., 2018*), one is due to a relatively low temperature (283 K) as well as the presence of glycerol (20% v/v; *Bafaro et al., 2019*), and two can be explained by tertiary structure formation [a folded domain (*Maiti et al., 2019*) or molten globule (*Patel et al., 2014*)]. A simple reason for higher $R_2$ values at lower temperatures is the higher water viscosity, resulting in a slowdown in molecular tumbling; a similar effect is achieved by adding glycerol. In some cases, $R_2$ was measured at both low and room temperatures (*Abyzov et al., 2016*; *Martin et al., 2020*). To a good approximation, the effect of lowering temperature is a uniform scaling of $R_2$ across the IDP

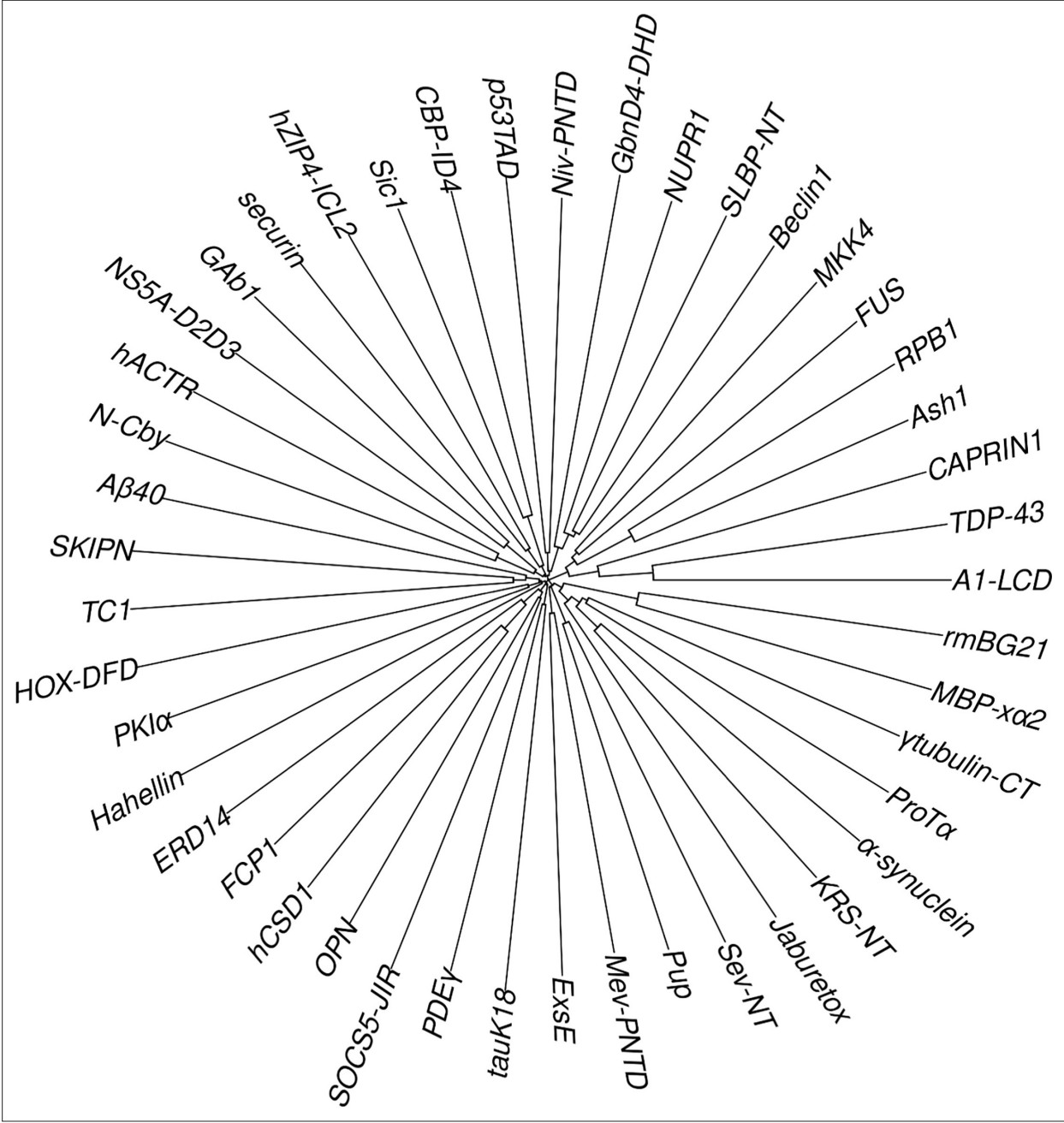

**Figure 1.** Clock-like tree plot showing lack of homology among the 45 IDPs. The level of homology between two sequences is measured by the distance from their convergence point to the center of the clock. The highest level of apparent identity is between A1-LCD and TDP-43, at 25%, but these two proteins differ in both secondary structure formation and $R_2$ characteristics. There is, however, a 20-residue overlap between the N-terminus of MBP-xα2 and the C-terminus of rmBG21.

sequence. For Sev-NT, downscaling of the $R_2$ values at 278 K by a factor of 2.0 brings them into close agreement with those at 298 K (**Figure 3—figure supplement 1C**), with a root-mean-square-deviation (RMSD) of 0.5 s$^{-1}$ among all the residues. Likewise, for A1-LCD, downscaling by a factor of 2.4 brings the $R_2$ values at 288 K into good match with those at 298 K (**Figure 3—figure supplement 1D**), with an RMSD of 0.4 s$^{-1}$. Because SeqDYN is concerned with the sequence dependence of $R_2$, a uniform scaling has no effect on model parameter or prediction; therefore mixing the data from different temperatures is justified. The same can be said about the different magnetic fields in acquiring the $R_2$ data (**Table 1** and **Figure 3—figure supplement 1E**). Increasing the magnetic field raises $R_2$ values,

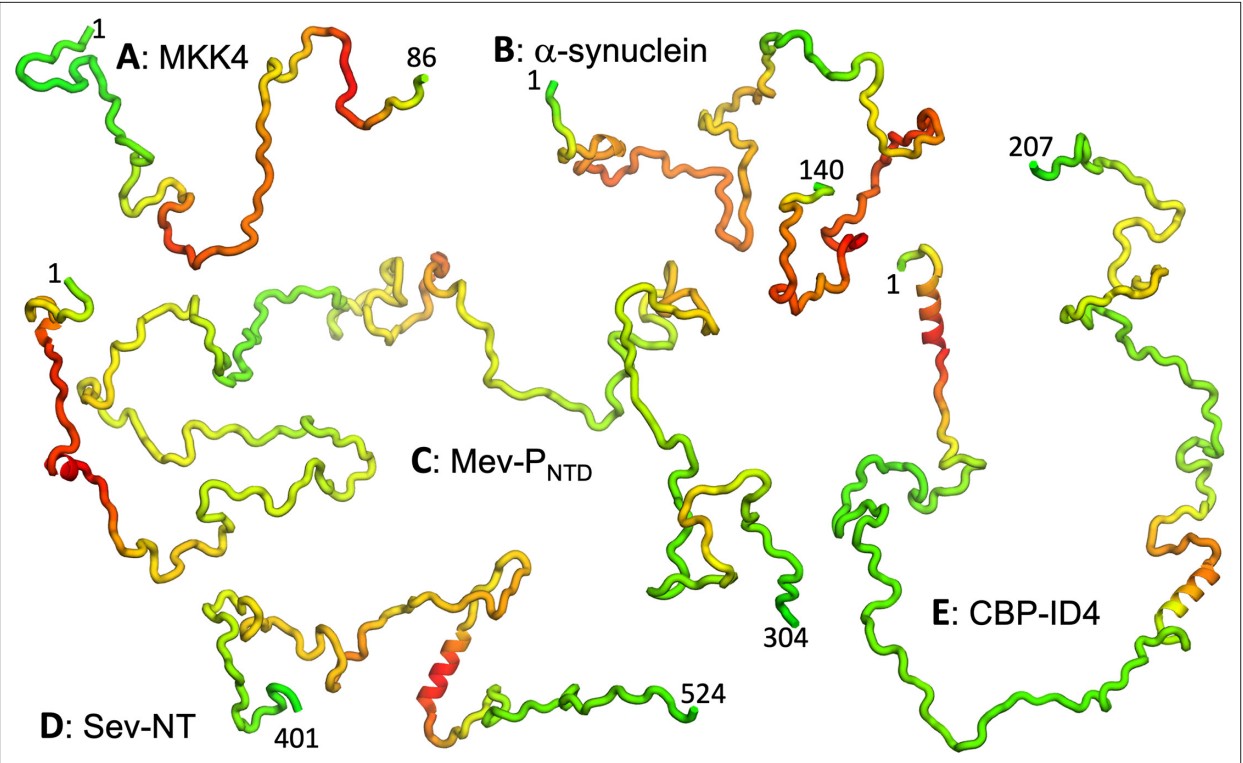

**Figure 2.** Representative conformations of five IDPs. (**A–E**) MKK4, α-synuclein, Mev-$P_{NTD}$, Sev-NT, and CBP-ID4. Conformations were initially generated using TraDES (http://trades.blueprint.org; *Feldman and Hogue, 2002*), selected to have radius of gyration close to predicted by a scaling function $R_g = 2.54N^{0.522}$ (Å) (*Bernadó and Blackledge, 2009*). Conformations for residues predicted as helical by PsiPred plus filtering were replaced by an ideal helix. Finally residues are colored according to a scheme ranging from green for low predicted $R_2$ to red for high predicted $R_2$.

and the effect is also approximated well by a uniform scaling (*Abyzov et al., 2016*; *Conicella et al., 2016*; *Janke et al., 2018*).

One measure on the level of sequence dependence of $R_2$ is the standard deviation, $\sigma_{R_2}$, calculated among the residues of an IDP. Among the training set, the $R_2$ values of 30 IDPs have moderate sequence variations, with $\sigma_{R_2}$ ranging from 0.5 to 1.5 s$^{-1}$ (*Table 1*); the histogram of $\sigma_{R_2}$ calculated for the entire training set peaks around 0.75 s$^{-1}$ (*Figure 3A*). There is a moderate correlation between $\sigma_{R_2}$ and $\bar{R}_2$ (*Figure 3A*, inset), reflecting in part the fact that $\sigma_{R_2}$ can be raised simply by a uniform upscaling, for example as a result of lowering temperature. Still, only two of the five IDPs with high $\bar{R}_2$ attributable to lower temperature or presence of glycerol are among the seven IDPs with high sequence variations ($\sigma_{R_2}>2$ s$^{-1}$). Therefore, the sequence variation of $R_2$ as captured by $\sigma_{R_2}$ manifests mostly the intrinsic effect of the IDP sequence, not the influence of external factors such as temperature or magnetic field strength. The mean $\sigma_{R_2}$ value among the training set is 1.24 s$^{-1}$.

One way to eliminate the influence of external factors is to scale the $R_2$ values of each IDP by its $\bar{R}_2$; we refer to the results as scaled $R_2$, or $sR_2$. We then pooled the $sR_2$ values for all residues in the training set, and separated them according to amino-acid types. The amino acid type-specific mean $sR_2$ values, or $msR_2$, are displayed in *Figure 3B*. The seven amino acids with the highest $msR_2$ in descending order are Trp, Arg, Tyr, Phe, Ile, His, and Leu. The presence of all the four aromatic amino acids in this "high-end" group immediately suggests π-π stacking as important for raising $msR_2$; the presence of Arg further implicates cation-π interactions. In the other extreme, the seven amino acids with the lowest $msR_2$ in ascending order are Gly, Cys, Val, Asp, Ser, Thr, and Asn. Gly is well-known as a flexible residue; it is also interesting that all the four amino acids with short polar sidechains are found in this "low-end" group. Pro has an excessively low $msR_2$ [with data from only two IDPs (*Murrali et al., 2018*; *Wong et al., 2020*)], but that is due to the absence of an amide proton.

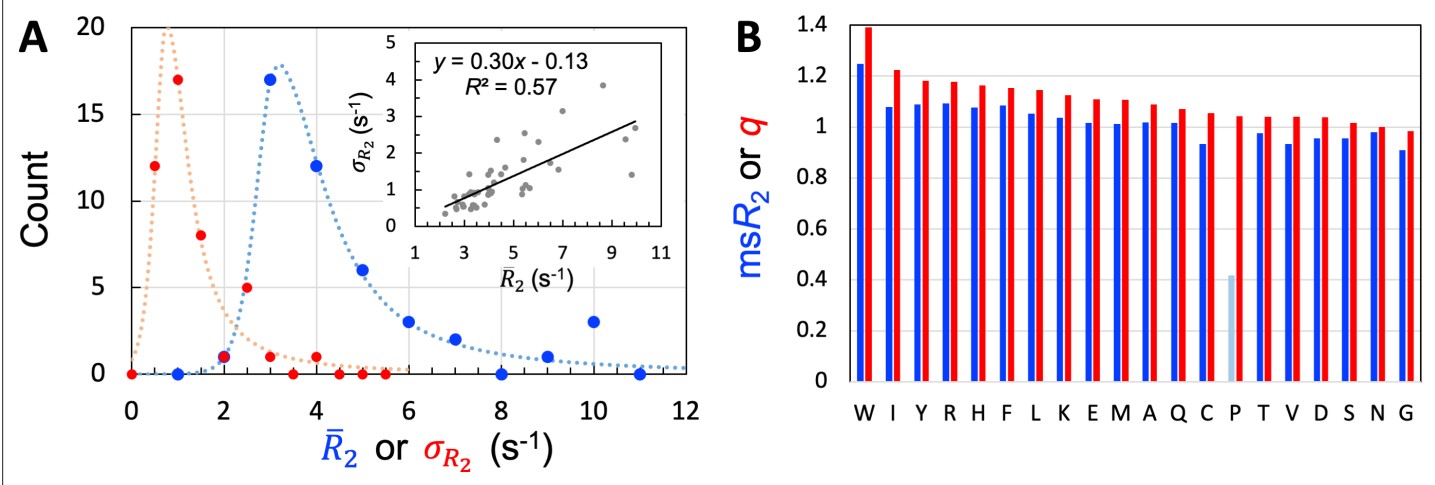

**Figure 3.** Properties of the 45 IDPs in the training set. (**A**) Histograms of means and standard deviations, calculated for individual proteins. Curves are drawn to guide the eye. Inset: correlation between $\bar{R}_2$ and $\sigma_{R_2}$. (**B**) Experimental mean scaled $R_2$ ($msR_2$) and SeqDYN $q$ parameters, for the 20 types of amino acids. Note that Pro residues have low $msR_2$ for the lack of backbone amide proton. Amino acids are in descending order of $q$.

The online version of this article includes the following source data and figure supplement(s) for figure 3:

**Source data 1.** Source data for *Figure 3*.

**Figure supplement 1.** Possible effects of sequence length, temperature, and magnetic field on $R_2$.

**Figure supplement 1—source data 1.** Source data for *Figure 3—figure supplement 1*.

## The SeqDYN model and parameters

The null model is to assume a uniform $R_2$ for all the residues in an IDP. The root-mean-square-error (RMSE) of the null model is equal to the standard deviation, $\sigma_{R_2}$, of the measured $R_2$ values. The mean RMSE, $\overline{RMSE}$, of the null model, equal to 1.24 s$^{-1}$ for the training set, serves as the upper bound for evaluating the errors of $R_2$ predictors. The next improvement is a one-residue predictor, where first each residue (with index $n$) assumes its amino acid-specific mean $sR_2$ ($msR_2$) and then a uniform scaling factor $\Upsilon$ is applied:

$$R_2(n) = \Upsilon \cdot msR_2(n) \qquad (1)$$

This one-residue model does only minutely better than the null model, with a $\overline{RMSE}$ of 1.22 s$^{-1}$.

In SeqDYN, we account for the influence of neighboring residues. Specifically, each residue $i$ contributes a factor $f(i;n)$ to the $R_2$ value of residue $n$. Therefore,

$$R_2(n) = \Upsilon \prod_{i=1}^{N} f(i;n) \qquad (2a)$$

where $N$ is the total number of residues in the IDP. The contributing factor depends on the sequence distance $s = |i - n|$ and the amino-acid type of residue $i$:

$$f(i;n) = 1 + \frac{q(i) - 1}{1 + bs^2} \qquad (2b)$$

There are 21 global parameters. The first 20 are the $q$ values, one for each of the 20 types of amino acids; the last parameter is $b$, appearing in the Lorentzian form of the sequence-distance dependence. We define the correlation length, $L_{corr}$, as the sequence distance at which the contributing factor is midway between the values at $s = 0$ and $\infty$. It is easy to verify that $L_{corr} = b^{-1/2}$. Note that the single-residue model can be seen as a special case of SeqDYN, with $L_{corr}$ set to 0 and $q$ set to $msR_2$.

The functional forms of *Equation 2a* and *Equation 2b* were adapted from *Li et al., 2020*; we also used them for predicting residue-specific membrane association propensities of IDPs (*Qin et al., 2022*). In these previous applications, a linear term was also present in the denominator of *Equation 2b*. In our initial training of SeqDYN, the coefficient of the linear term always converged to near zero. We thus eliminated the linear term. In addition to the Lorentzian form, we also tested a Gaussian form

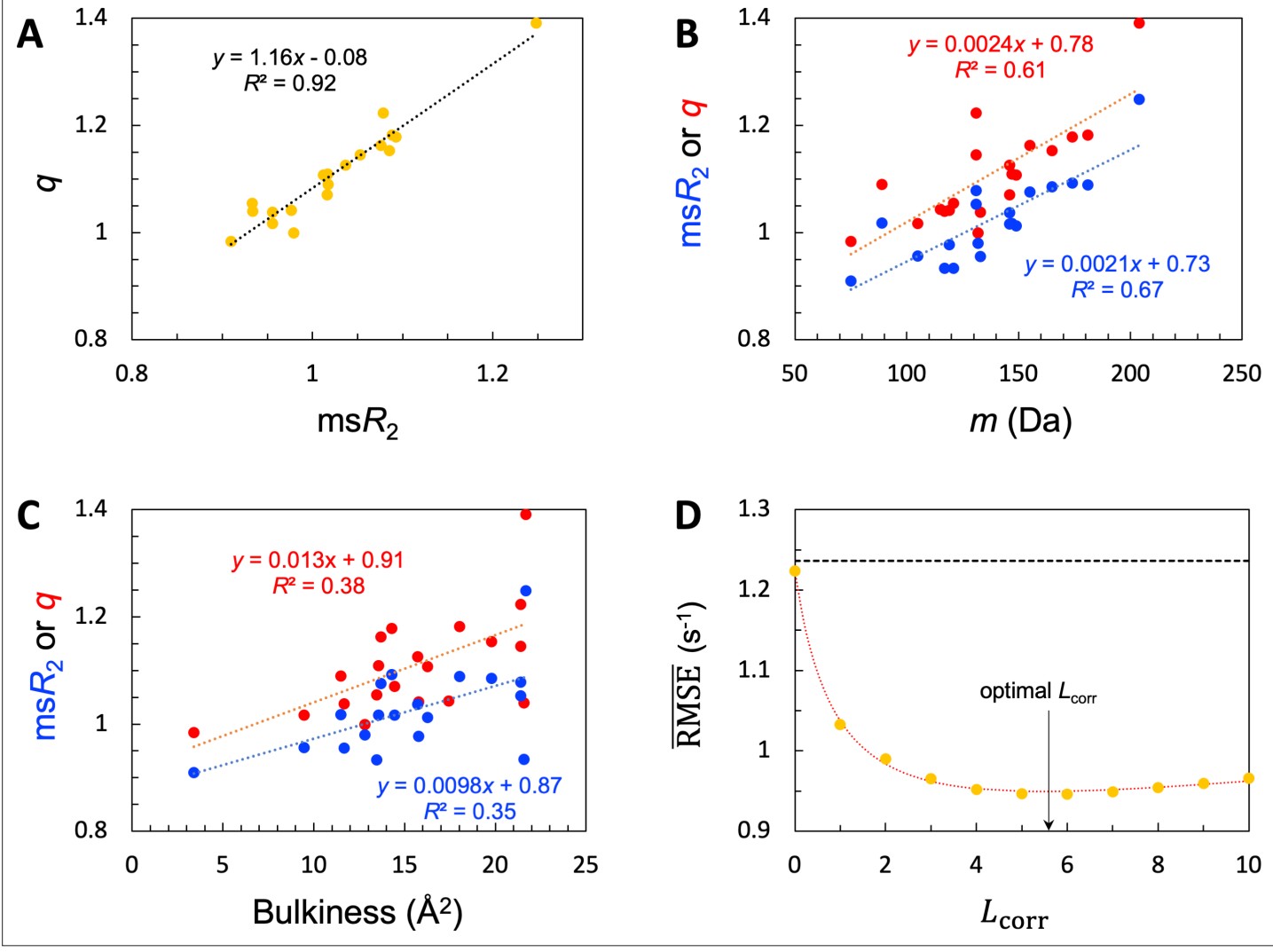

**Figure 4.** SeqDYN model parameters. (**A**) Correlation between $msR_2$ and $q$. The values are also displayed as bars in **Figure 3B**. (**B**) Correlation of $msR_2$ and $q$ with amino-acid molecular mass. (**C**) Correlation of $msR_2$ and $q$ with bulkiness. (**D**) The optimal correlation length and deterioration of SeqDYN prediction as the correlation length is moved away from the optimal value.

The online version of this article includes the following source data and figure supplement(s) for figure 4:

**Source data 1.** Source data for **Figure 4**.

**Figure supplement 1.** T-test of on the $q$ parameters for pairs of amino acids.

**Figure supplement 1—source data 1.** Source data for **Figure 4—figure supplement 1**.

for the sequence-distance dependence and found somewhat worse performance. The more gradual attenuation of the Lorentzian form with increasing sequence distance evidently provides an overall better model for the $R_2$ data in the entire training set. Others (**Cho et al., 2007**; **Sekiyama et al., 2022**; **Delaforge et al., 2018**) have modeled $R_2$ as the average of some parameters over a window; a window has an extremely abrupt sequence-distance dependence (1 for $s < L_{corr}$ and 0 for $s > L_{corr}$).

We parametrized the SeqDYN model represented by **Equation 2a** and **Equation 2b** on the training set of 45 IDPs. In addition to the 21 global parameters noted above, there are also 45 local parameters, namely one uniform scaling factor ($\Upsilon$) per IDP. The parameter values were selected to minimize the sum of the mean-square-errors for the IDPs in the training set, calculated on $R_2$ data for a total of 3924 residues. We excluded the 42 Pro residues in the training set because, as already noted, their $R_2$ values are lower for chemical reasons. We will present validation and test results below, but first let us look at the parameter values.

The $q$ values are displayed in **Figure 3B** alongside $msR_2$. In descending order, the seven amino acids with the highest $q$ values are Trp, Ile, Tyr, Arg, His, Phe, and Leu. These are exactly the same amino acids in the high-end group for $msR_2$, though their order there is somewhat different. In ascending order, the seven amino acids (excluding Pro) with the lowest $q$ values are Gly, Asn, Ser, Asp, Val, Thr, and Cys. The composition of the low-end group is also identical to that for $msR_2$. The $q$ values thus also suggest that π-π and cation-π interactions in local sequences may raise $R_2$, whereas Gly and short-polar residues may lower $R_2$.

Given the common amino acids at both the high and low ends for $msR_2$ and $q$, it is not surprising that these two properties exhibit a strong correlation, with a coefficient of determination ($R^2$; excluding Pro) at 0.92 (**Figure 4A**). Also, because the high-end group contains the largest amino acids (e.g. Trp and Tyr) whereas the low-end group contains the smallest amino acids (e.g. Gly and Ser), we anticipated some correlation of $msR_2$ and $q$ with amino-acid size. We measure the latter property by the molecular mass ($m$). As shown in **Figure 4B**, both $msR_2$ and $q$ indeed show a medium correlation with $m$, with $R^2=0.67$ (excluding Pro) and 0.61, respectively. A bulkiness parameter was proposed as an indicator of sequence-dependent backbone dynamics of IDPs (**Cho et al., 2007**; **Delaforge et al., 2018**). Bulkiness was defined as the sidechain volume-to-length ratio, and identified amino acids with aromatic or branched aliphatic sidechains as bulky (**Zimmerman et al., 1968**). We found only modest correlations between either $msR_2$ or $q$ and bulkiness, with $R^2$ just below 0.4 (**Figure 4C**).

The optimized value of $b$ is $3.164 \times 10^{-2}$, corresponding to an $L_{corr}$ of 5.6 residues. The resulting optimized $R\overline{M}SE$ is 0.95 s$^{-1}$, a clear improvement over the value 1.24 s$^{-1}$ of the null model. To check the sensitivity of prediction accuracy to $b$, we set $b$ to values corresponding to $L_{corr}$ = 0, 1, 2,…, and retrained SeqDYN for $b$ fixed at each value (**Figure 4D**). Note that the null-model $R\overline{M}SE$, 1.24 s$^{-1}$, sets an upper bound. This upper bound is slowly reached when $L_{corr}$ is increased from the optimal value. In the opposite direction, when $L_{corr}$ is decreased from the optimal value, $R\overline{M}SE$ rises quickly, reaching 1.22 s$^{-1}$ at $L_{corr}$ = 0. The latter $R\overline{M}SE$ is the same as that of the single-residue model. Lastly we note that there is a strong correlation between the uniform scaling factors and $\bar{R}_2$ values among the 45 IDPs ($R^2=0.77$), as to be expected. For 39 of the 45 IDPs, Υ values fall in the range of 0.8–2.0 s$^{-1}$.

As presented next, we evaluate the performance of SeqDYN by leave-one-out cross validation, where each IDP in turn was left out of the training set and the model was trained on the remaining 44 IDPs to predict $R_2$ for the IDP that was left out. The parameters from the leave-one-out (also known as jackknife) training sessions allow us to assess the potential bias of the training set. For this purpose, we compare the values of the 21 global parameters, either from the full training set or from taking the averages of the jackknife training sessions. For each of the $q$ parameters, the values from these two methods differ only in the fourth digit; for example for Leu, they are both 1.1447 from full training and from jackknife training. The values for $b$ are $3.164\times10^{-2}$ from full training as stated above and $3.163\times10^{-2}$ from jackknife training. The close agreement in parameter values between full training and jackknife training suggests no significant bias in the training set.

Another question of interest is whether the difference between the $q$ parameters of two amino acids is statistically significant. To answer this question, we carried out fivefold cross-validation training, resulting in five independent estimates for each parameter. For example, the mean ±standard deviation of the $q$ parameter is 1.1405 ± 0.0066 for Leu and 1.2174 ± 0.0211 for Ile. A t-test shows that their difference is extremely statistically significant ($P<0.0001$). In contrast, the difference between Leu and Phe ($q=1.1552 ± 0.0304$) is not significant. t-test results for other pairs of amino acids are found in **Figure 4—figure supplement 1**.

## Validation of SeqDYN predictions

We now present leave-one-out cross-validation results. We denote the RMSE of the $R_2$ prediction for the left-out IDP as RMSE(–1). As expected, RMSE(–1) is higher than the RMSE obtained with the IDP kept in the training set, but the increases are generally slight. Specifically, all but eight of the IDPs have increases <0.1 s$^{-1}$; the largest increase is 0.35 s$^{-1}$, for CBP-ID4. The mean RMSE(–1), or $R\overline{M}SE(-1)$, for the 45 IDPs is increased by 0.05 s$^{-1}$ over $R\overline{M}SE$, to 1.00 s$^{-1}$. The latter value is still a distinct improvement over the mean RMSE 1.24 s$^{-1}$ of the null model. The histogram of RMSE(–1) for the 45 IDPs is shown in **Figure 5A**. It peaks at 0.5 s$^{-1}$, which is a substantial downshift from the corresponding peak at 0.75 s$^{-1}$ for $\sigma_{R_2}$ (**Figure 3A**). Thirty-four of the 45 IDPs have RMSE(–1) values lower than the corresponding $\sigma_{R_2}$.

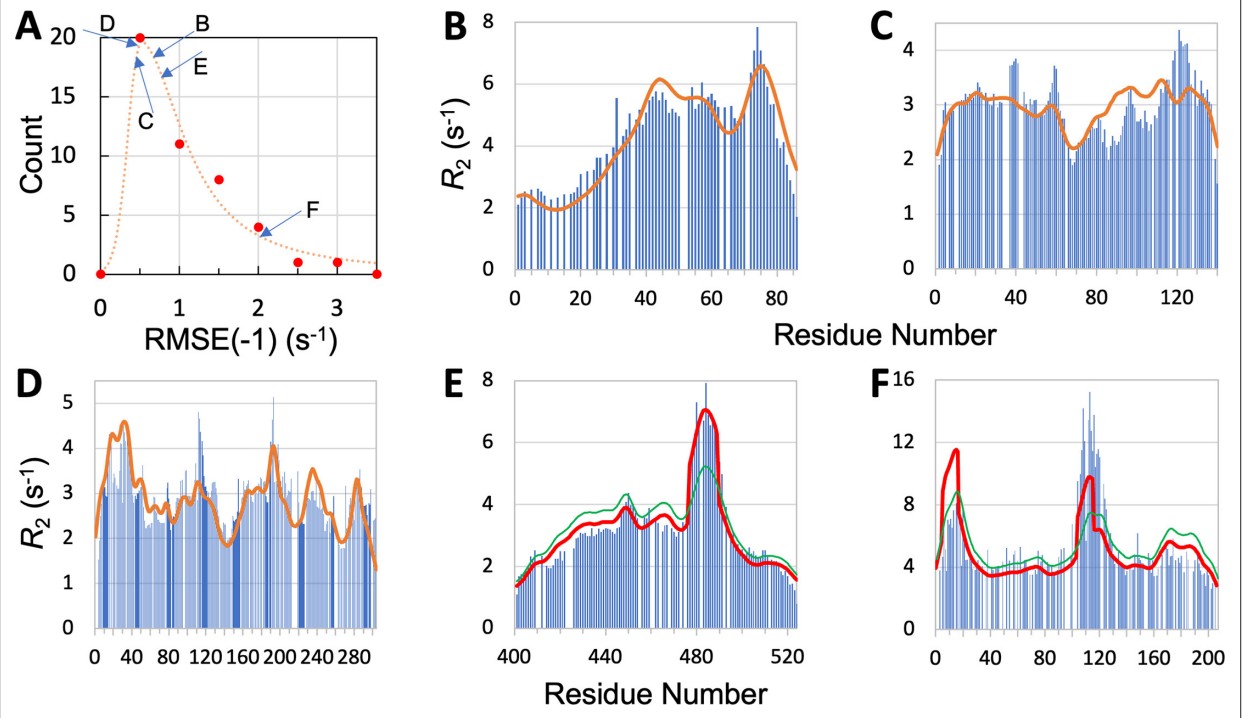

**Figure 5.** Quality of SeqDYN predictions. (**A**) Histogram of RMSE(–1). Letters indicate RMSE(–1) values of the IDPs to be presented in panels (**B–F**). (**B–F**) Measured (bars) and predicted (curves) $R_2$ profiles for MKK4, α-synuclein, Mev-$P_{NTD}$, Sev-NT, and CBP-ID4. In (**E**) and (**F**), green curves are SeqDYN predictions and red curves are obtained after a helix boost.

The online version of this article includes the following source data and figure supplement(s) for figure 5:

**Source data 1.** Source data for *Figure 5*.

**Figure supplement 1.** Close reproduction (curve) of the measured $R_2$ profile (bars) of CBP-ID4 when that set of data alone was used to parameterize SeqDYN.

**Figure supplement 1—source data 1.** Source data for *Figure 5—figure supplement 1*.

To further illustrate the performance of SeqDYN, we present the comparison of predicted and measured $R_2$ values for five IDPs: MKK4, α-synuclein, Mev-$P_{NTD}$, Sev-NT, and CBP-ID4 (*Figure 5B–F*). A simple common feature is the falloff of $R_2$ at the N- and C-termini, resulting from missing upstream or downstream residues that otherwise would be coupled to the terminal residues, as first recognized by *Schwalbe et al., 1997*. Representative conformations of the five IDPs are displayed in *Figure 2*, with residues colored according to the predicted $R_2$ values. For four of these IDPs, the RMSE(–1) values range from 0.44 to 0.76 $s^{-1}$ and are scattered around the peak of the histogram, while the RMSE(–1) for the fifth IDP, namely CBP-ID4, the RMSE(–1) value is 2.01 $s^{-1}$ and falls on the tail of the histogram (*Figure 5A*). *Figure 5B* displays the measured and predicted $R_2$ for MKK4. SeqDYN correctly predicts higher $R_2$ values in the second half of the sequence than in the first half. It even correctly predicts the peak around residue Arg75. The sequence in this region is $H_{72}$IERLRTH$_{79}$; six of these eight residues belong to the high-end group. In contrast, the lowest $R_2$ values occur in the sequence $S_7$GGGGS-GGGSGSG$_{19}$, comprising entirely of two amino acids in the low-end group.

$R_2$ values for α-synuclein are shown in *Figure 5C*. Here, SeqDYN correctly predicts higher $R_2$ near the C-terminus and a dip around Gly68. However, it misses the $R_2$ peaks around Tyr39 and Asp121. MD simulations *Dey et al., 2022* have found that these $R_2$ peaks can be explained by a combination of secondary structure formation (β-sheet around Tyr39 and polyproline II helix around Asp121) and local (between Tyr39 and Ser42) or long-range (between Asp121 and Lys96) interactions. SeqDYN cannot account for long-range interactions (e.g. between β-strands and between Asp121 and Lys96). *Figure 5D* shows that SeqDYN gives excellent $R_2$ predictions for Mev-$P_{NTD}$. It correctly predicts the high peaks around Arg17, Glu31, Leu193, and lower peaks around Arg235 and Trp285, but does underpredict the narrow peak around Tyr113.

The overall $R_2$ profile of Sev-NT is predicted well by SeqDYN, but the peak in the long helical region (residues 478–491) is severely underestimated (green curve in *Figure 5E*). A similar situation occurs for CBP-ID4, where the peak in the second long helical region (around Glu113) is underpredicted (green curve in *Figure 5F*). While the measured $R_2$ exhibits a higher peak in the second helical region than in the first helical region (around Arg16), the opposite is predicted by SeqDYN. When the $R_2$ data were included in the training set (i.e., full training), the second peak is higher than the first one, but that is not a real prediction because the $R_2$ data themselves were used for training the model. It merely means that the SeqDYN functions can be parameterized to produce any prescribed $R_2$ profile along the sequence. Indeed, when the $R_2$ data of CBP-ID4 alone were used to parameterize SeqDYN, the measured $R_2$ profile is closely reproduced (*Figure 5—figure supplement 1*). The reversal in $R_2$ peak heights between the two helical regions is the reason for the aforementioned unusual increase in RMSE when CBP-ID4 was left out of the training set.

## R2 boost in long helical regions

It is apparent that SeqDYN underestimates the $R_2$ of stable long helices. Transient short helices does not seem to be a problem, since these are present, for example in Mev-P$_{NTD}$, where transient helix formation in the first 37 residues and between residues 189–198 (*Milles et al., 2018*) coincides with $R_2$ peaks that are correctly predicted by SeqDYN. SeqDYN can treat coupling between residues within

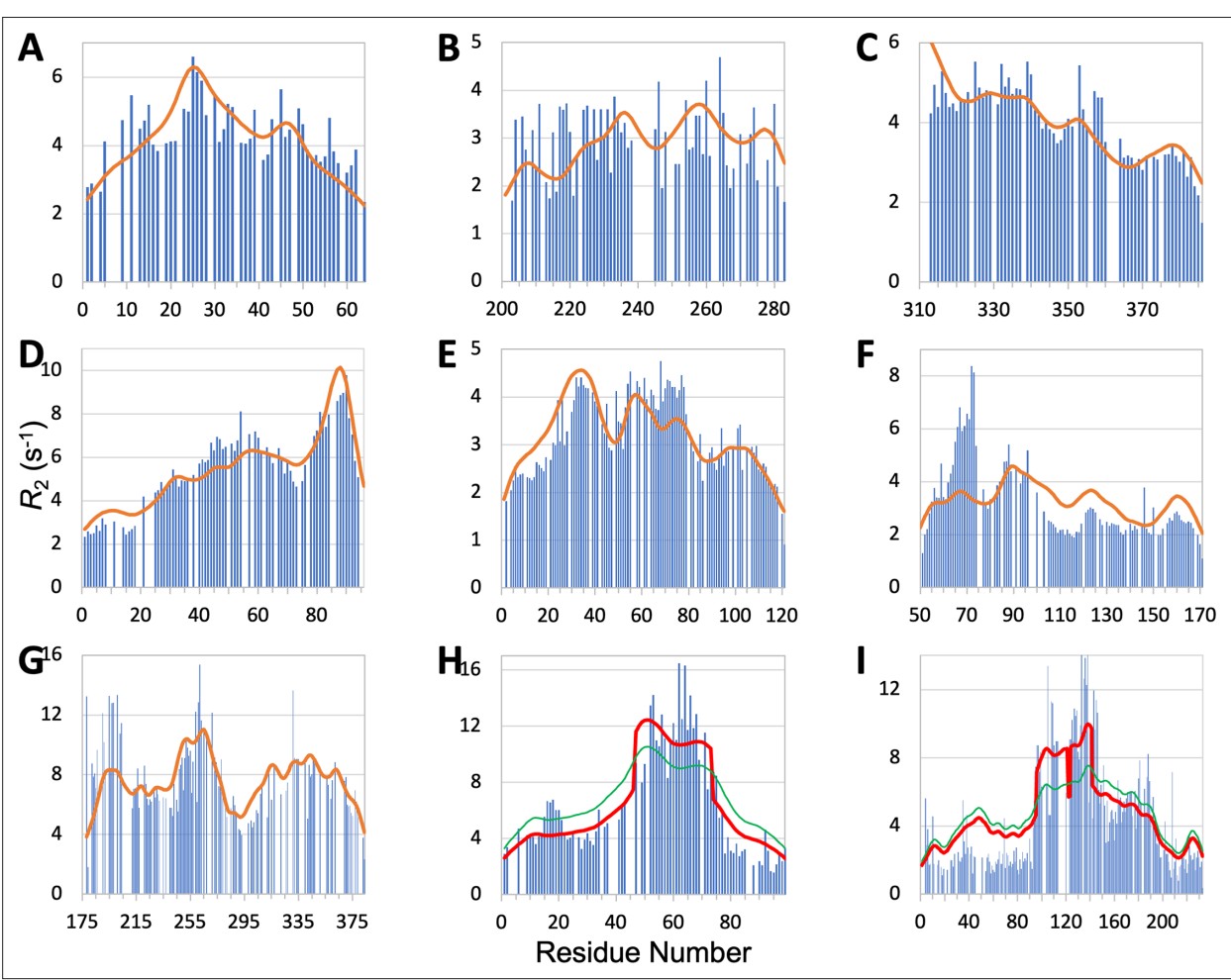

**Figure 6.** Measured (bars) and predicted (curves) $R_2$ profiles for ChiZ N-terminal region, TIA1 prion-like domain, Pdx1 C-terminal region, synaptobrevin-2, α-endosulfine, YAP, AMOTL1, FtsQ, and CAHS-8. In (**C**), $R_2$ does not fall off at the N-terminus because the sequence is preceded by an expression tag MGSSHHHHHHHHHHHHHHS. In (**H**) and (**I**), green curves are SeqDYN predictions and red curves are obtained after a helix boost.

The online version of this article includes the following source data for figure 6:

**Source data 1.** Source data for *Figure 6*.

the correlation length of 5.6 residues, but a much longer helix would tumble more slowly than implied by an $L_{corr}$ of 5.6, and thus it makes sense that SeqDYN would underestimate $R_2$ in that case.

Our solution then is to apply a boost factor to the long helical region. To do so, we have to know whether an IDP does form long helices and if so what the constituent residues are. Secondary structure predictors tend to overpredict α-helices and β-strands for IDPs, as they are trained on structured proteins. One way to counter that tendency is to make the criteria for α-helices and β-strands stricter. We found that, by filtering PsiPred (http://bioinf.cs.ucl.ac.uk/psipred; *McGuffin et al., 2000*) helix propensity scores ($p_{Hlx}$) with a very high cutoff of 0.99, the surviving helix predictions usually correspond well with residues identified by NMR as having high helix propensities. For example, for Mev-P$_{NTD}$, PsiPred plus filtering predicts residues 14–17, 28–33, and 191–193 as helical; all of them are in regions that form transient helices according to chemical shifts (*Milles et al., 2018*). Likewise long helices are also correctly predicted for Sev-NT (residues 477–489) and CBP-ID4 (residues 6–17 and 105–116; *Jensen et al., 2008*; *Piai et al., 2016*).

We apply a boost factor, $B_{Hlx}$, to helices with a threshold length of 12:

$$B_{Hlx} = 1 + \alpha p_{Hlx} \Theta \left( p_{Hlx} \geq 0.99; L_{Hlx} \geq 12 \right) \tag{3}$$

The $\Theta$ function is 1 if the helix propensity score is above the filtering cutoff and the helix length ($L_{Hlx}$) is above the threshold, and 0 otherwise With a boost amplitude $\alpha$ at 0.5, the boosted SeqDYN prediction for Sev-NT reaches excellent agreement with the measured $R_2$ (*Figure 5E*, red curve). The RMSE(–1) is reduced from 0.76 s$^{-1}$ to 0.38 s$^{-1}$ upon boosting. Applying the same helix boost to CBP-ID4 also results in a modest reduction in RMSE(–1), from 2.01 to 1.90 s$^{-1}$ (*Figure 5F*, red curve). The only other IDP for which PsiPred plus filtering predicts a long helix is the N-terminal region of lysyl-tRNA synthetase (KRS-NT). The authors who studied this protein did not report on secondary structure (*Cho et al., 2014*), but feeding their reported chemical shifts to the TALOS +server (https://spin.niddk.nih.gov/bax/nmrserver/talos/; *Shen et al., 2009*) found only short stretches of residues that fall into the helical region of the Ramachandran map. The SeqDYN prediction for KRS-NT is already good [RMSE(–1)=0.83 s$^{-1}$]; applying a helix boost would deteriorate the RMSE(−1) to 1.16 s$^{-1}$.

## Further test on a set of nine IDPs

We have reserved nine IDPs for testing SeqDYN (parameterized on the training set of 45 IDPs). The level of disorder in these test proteins also spans the full range, from absence of secondary structures [ChiZ N-terminal region (*Hicks et al., 2020*), Pdx1 C-terminal region (*Cook et al., 2019*), and TIA-1 prion-like domain (*Sekiyama et al., 2022*)] to presence of transient short helices [synaptobrevin-2 (*Lakomek et al., 2019*), α-endosulfine (*Thapa et al., 2022*), YAP (*Feichtinger et al., 2022*), angiomotin-like 1 (AMOTL1) (*Vogel et al., 2022*)] to formation of stable long helices [FtsQ *Smrt et al., 2023* and CAHS-8 *Malki et al., 2022*]. For eight of the nine test IDPs, the RMSEs of SeqDYN predictions are lower than the experimental $\sigma_{R_2}$ values, by an average of 0.66 s$^{-1}$. For the ninth IDP (Pdx1), the SeqDYN RMSE is slightly higher, by 0.06 s$^{-1}$, than the experimental $\sigma_{R_2}$. Together, the nine test IDPs have a mean RMSE of 1.13 s$^{-1}$, close to the $\overline{RMSE}(-1)$ of 1.00 s$^{-1}$ for the training set in the leave-one-out cross-validation.

The comparison of predicted and measured $R_2$ profiles along the sequence is presented in *Figure 6A–I*. For ChiZ, SeqDYN correctly predicts the major peak around Arg25 and the minor peak around Arg46 (*Figure 6A*). The $R_2$ profile of Pdx1 is largely featureless, except for a dip around Gly216, which is correctly predicted by SeqDYN (*Figure 6B*). Correct prediction is also obtained for the higher $R_2$ in the first half of TIA-1 prion-like domain than in the second half (*Figure 6C*). SeqDYN gives an excellent prediction for synaptobrevin-2, including a linear increase up to Arg56 and the major peak around Trp89 (*Figure 6D*).

The prediction is also very good for α-endosulfine, including elevated $R_2$ around Glu34, which coincides with the presence of a transient helix, and depressed $R_2$ in the last 40 residues (*Figure 6E*). The only miss is an underprediction for the peak around Lys74. SeqDYN also predicts well the overall shape of the $R_2$ profile for YAP, including peaks around Asn70, Leu91, Arg124, and Arg161, but severely underestimates the peak height around Asn70 (*Figure 6F*). NOE signals indicate contacts between Met86, Leu91, Fhe95, and Fhe96 (*Feichtinger et al., 2022*); evidently this type of local contacts is captured well by SeqDYN. The $R_2$ elevation around Asn70 is mostly due to helix formation: residues 61–74 have helix propensities up to 40% (*Feichtinger et al., 2022*). PsiPred predicts helix for

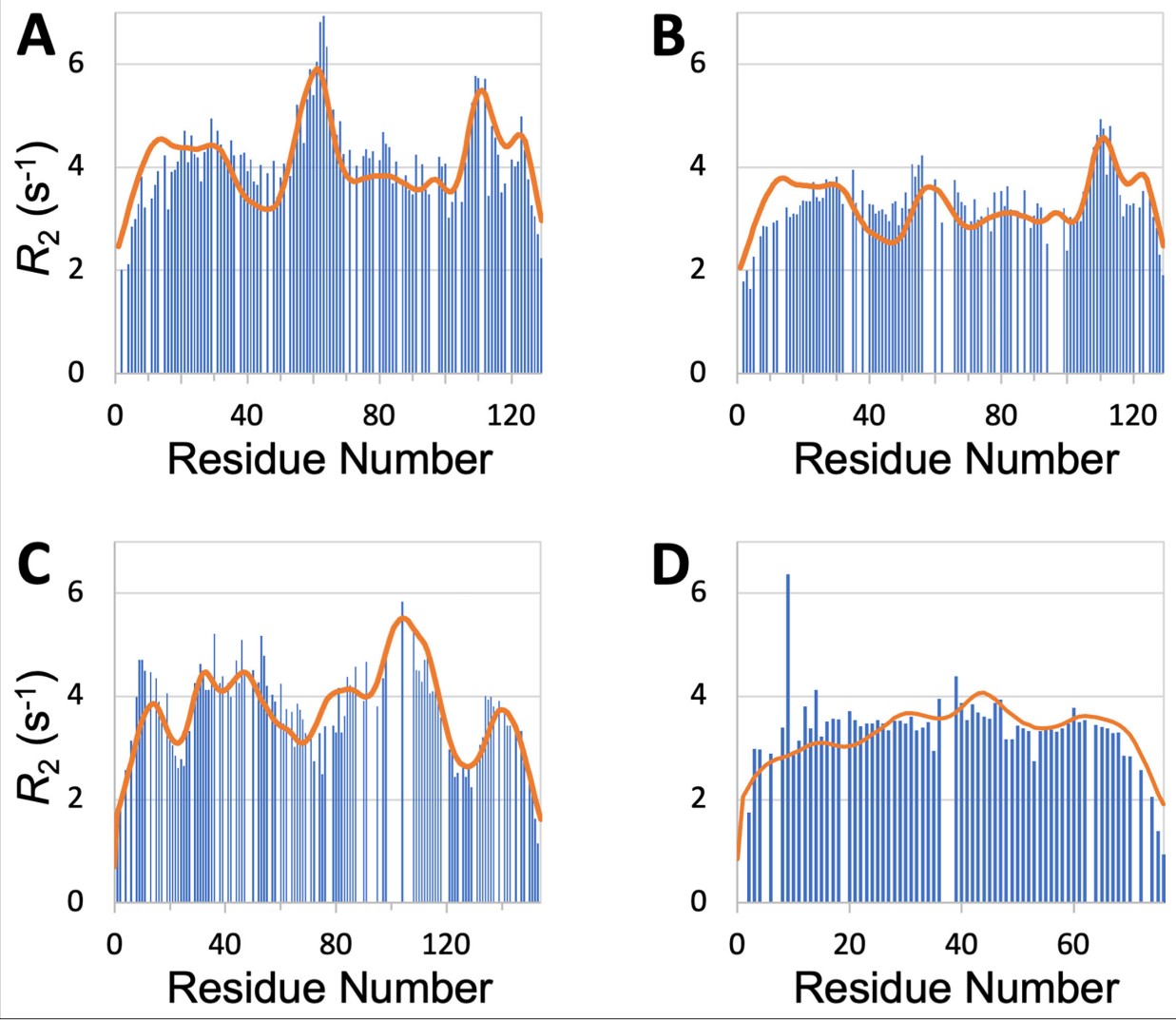

**Figure 7.** $R_2$ profiles predicted (curves) by SeqDYN show close agreement with those measured (bars) on structured proteins in the unfolded state. (**A**) Wild-type lysozyme (8 M urea; pH 2; cysteine-methylated). (**B**) Lysozyme with Trp62 to Gly mutation (pH 2). Methylated cysteines were treated as Ala in the SeqDYN predictions. (**C**) Apomyoglogin (8 M urea; pH 2.3). (**D**) Ubiquitin (8 M urea; pH 2).

The online version of this article includes the following source data for figure 7:

**Source data 1.** Source data for *Figure 7*.

residues 62–73, but only residues 65–68 survive the filtering that we impose, resulting in a helix that is too short to apply a helix boost. The prediction for AMOTL1 is mostly satisfactory, including peaks around Phe200 and Arg264 and a significant dip around Gly292 (*Figure 6G*). However, whereas the two peaks have approximately equal heights in the measured $R_2$ profile, the predicted peak height around Phe200 is too low. SCSs indicate helix propensity around both $R_2$ peaks (*Vogel et al., 2022*). PsiPred also predicts helix in both regions, but only five and two residues, respectively, survive after filtering, and are too short for applying a helix boost.

For FtsQ, SeqDYN correctly predicts elevated $R_2$ for the long helix [residues 46–74 *Smrt et al., 2023*] but underestimates the magnitude (RMSE = 2.32 s⁻¹; green curve in *Figure 6H*). PsiPred plus filtering predicts a long helix formed by residues 47–73. Applying the helix boost substantially improves the agreement with the measured $R_2$, with RMSE reducing to 1.71 s⁻¹ (red curve in *Figure 6H*). SeqDYN also gives a qualitatively correct $R_2$ profile for CAHS-8, with higher $R_2$ for the middle section (residues 95–190; RMSE = 2.36 s⁻¹; green curve in *Figure 6I*). However, it misses the extra elevation in $R_2$ for the first half of the middle section (residues 95–145). According to SCS, the first and second halves have helix propensities of 60% and 30%, respectively (*Malki et al., 2022*). PsiPred plus filtering predicts

helices for residues 96–121, 124–141, 169–173, and 179–189. Only the first two helices, both in the first half of the middle section, are considered long according to our threshold. Once again, applying the helix boost leads to marked improvement in the predicted in $R_2$, with RMSE reducing to 1.92 s$^{-1}$ (red curve in *Figure 6I*).

## Inputting the sequences of structured proteins predicts $R_2$ in the unfolded state

SeqDYN is trained on IDPs, what if we feed it with the sequence of a structured protein? The prediction using the sequence of hen egg white lysozyme, a well-studied single-domain protein, is displayed in *Figure 7A*. It shows remarkable agreement with the $R_2$ profile measured by Klein-**Klein-Seetharaman et al., 2002** in the unfolded state (denatured by 8 M urea at pH 2 and reduced to break disulfide bridges), including a major peak around Trp62, a second peak around Trp111, and a third peak around Trp123. Klein-Seetharaman et al. mutated Trp62 to Gly and the major peak all but disappeared. This result is also precisely predicted by SeqDYN with the mutant sequence (*Figure 7B*).

SeqDYN also predicts well the $R_2$ profiles of other proteins in the unfolded state. For unfolded apomyoglobin (8 M urea; pH 2.3), **Schwarzinger et al., 2002** claimed that depressed $R_2$ corresponded to stretches of small amino acids (Gly and Ala), whereas elevated corresponded to local hydrophobic interactions. SeqDYN reproduces all the observed peaks and valleys in the $R_2$ profile (*Figure 7C*). The deepest valley indeed occurs over a Gly/Ala-rich stretch, $G_{125}ADAQGA_{131}$, but the highest peak occurs over a stretch, $I_{102}KYLEFI_{108}$, that contains both hydrophobic and charged residues, all of which are on the high end of the $q$ parameters (*Figure 3B*). The $R_2$ profile of unfolded ubiquitin (8 M urea; pH 2) is relatively flat, which **Wirmer et al., 2006** attributed to lack of residual secondary structure, based on the assumption that β-sheets (major elements of folded ubiquitin) are less resistant to denaturation than α-helices. SeqDYN predicts a relatively flat $R_2$ profile (*Figure 7D*), but the reason is that the ubiquitin sequence lacks a contiguous stretch of high-$q$ amino acids.

## Discussion

We have developed a powerful method, SeqDYN, that predicts the backbone amide transverse relaxation rates ($R_2$) of IDPs. The method is based on IDP sequences, is extremely fast, and available as a web server at https://zhougroup-uic.github.io/SeqDYNidp/ (**Qin and Zhou, 2024**). The excellent performance supports the notion that the ns-dynamics reported by $R_2$ is coded by the local sequence, comprising up to 6 residues on either side of a given residue. The amino-acid types that contribute the most to coupling within a local sequence are aromatic (Trp, Tyr, Phe, and His), Arg, and long branched aliphatic (Ile and Leu), suggesting the importance of π-π, cation-π, and hydrophobic interactions in raising $R_2$. These interactions are interrupted by Gly and amino acids with short polar sidechains (Ser, Thr, Asn, and Asp), leading to reduced $R_2$. Transient short helices produce moderate elevation in $R_2$, whereas stable long helices result in a big boost in $R_2$. Tertiary contacts can also raise $R_2$, but appear to be infrequent in most IDPs (**Dey et al., 2022**).

It is also possible that $R_2$ reported by backbone amide $^{15}$N relaxation (as is the case for most of the IDPs studied here) may not be particularly sensitive to exchange effects, which likely involve tertiary contact formation. For the D2 domain of p27$^{Kip1}$, the exchange contributions measured using $^{15}$N relaxation were small (<2.5 s$^{-1}$) but were as large as 25 s$^{-1}$ when measured by high-power $^1$H relaxation dispersion (**Ban et al., 2017**). This experiment measures the effective transverse relation rate, $R_{2,eff}$, over a range of effective radiofrequency $\omega_{eff}$. The exchange contribution is maximal for the value $R_{2,eff}^{low\omega_{eff}}$ at low $\omega_{eff}$ but is largely quenched for the value $R_{2,0}^{app}$ in the high-$\omega_{eff}$ limit. The SeqDYN prediction for this IDP matches much better with $R_{2,0}^{app}$ than with $R_{2,eff}^{low\omega_{eff}}$ (*Figure 8*). It is not clear whether this IDP is unique in forming persistent tertiary contacts that give rise to substantial exchange contributions or the $^1$H relaxation dispersion experiment is unique in reporting the exchange contributions. At the minimum, SeqDYN yields the exchange-free portion of the transverse relaxation rate, enabling easy identification of residues that potentially participate in tertiary contacts. For the D2 domain of p27$^{Kip1}$, SeqDYN correctly predicts the $R_{2,0}^{app}$ local maxima at W76 and Y88. It is these same two residues that show substantial exchange contributions and putatively participate in tertiary contact (**Ban et al., 2017**). Therefore local contacts may seed tertiary contacts. If $R_{2,eff}$ data with substantial

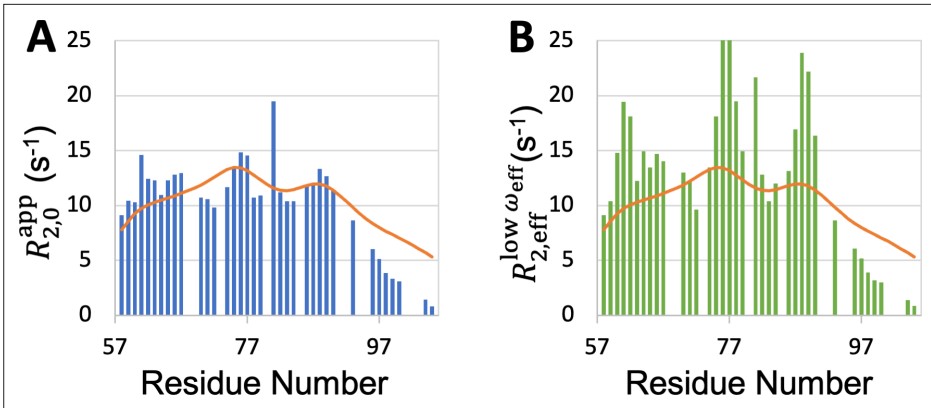

**Figure 8.** Comparison between SeqDYN prediction (curves) and effective transverse relaxation rate (bars) from [1]H dispersion relaxation experiment. (**A**) $R_{2,eff}$ in the high-$\omega_{eff}$ limit. (**B**) $R_{2,eff}$ at low $\omega_{eff}$.

The online version of this article includes the following source data for figure 8:

**Source data 1.** Source data for *Figure 8*.

exchange contributions become available for more IDPs, SeqDYN may be retrained to make predictions for IDPs forming persistent tertiary contacts.

The $q$ parameters, while introduced here to characterize the propensities of amino acids to participate in local interactions, appear to correlate with the tendencies of amino acids to drive liquid-liquid phase separation. Consistent with the rank order of $q$, Trp, Tyr, and Arg have been reported to be strong drivers of phase separation, Lys is a moderate driver, whereas Gly and Ser suppress phase separation (*Martin et al., 2020*; *Wong et al., 2020*; *Wang et al., 2018*). Recent measurements of the threshold concentration produced the following order for the propensity of phase separation by eight nonpolar amino acids in homotetrapeptides of the form XXssXX (ss: backbone disulfide bond): Trp > Phe > Leu>Met > Ile>Val > Ala>Pro (*Zhang et al., 2024*). This order is the same as that of the $q$ parameters, except that the $q$ values of Ile and Val are in the second and last places, respectively. Threshold concentrations of IDPs are now predicted reasonably well by coarse-grained simulations where each amino acid is modeled by a single bead with a Lennard-Jones diameter $d_0$ and a stickiness parameter $\lambda$ (*Tesei and Lindorff-Larsen, 2022*). Our $q$ parameter shows a good correlation ($R^2$=0.59) with the compound parameter $d_0^3\lambda$ (*Figure 9*). Therefore, the $q$ parameter may serve as a predictor for the tendency of an amino acid to drive phase separation. In essence, the same ability of an amino acid, for example Trp, to form interactions with neighboring residues of an IDP in the free state also applies when it comes to interactions with residues on neighboring chains in a dense phase.

Our method incorporates ideas from a number of previous efforts at describing $R_2$. The first serious effort was by *Schwalbe et al., 1997*, who accounted for contributions from neighboring residues as additive terms, instead of multiplicative factors as in SeqDYN. *Cho et al., 2007* and *Delaforge et al., 2018* used the running average of the bulkiness parameter over a window of five to nine residues as

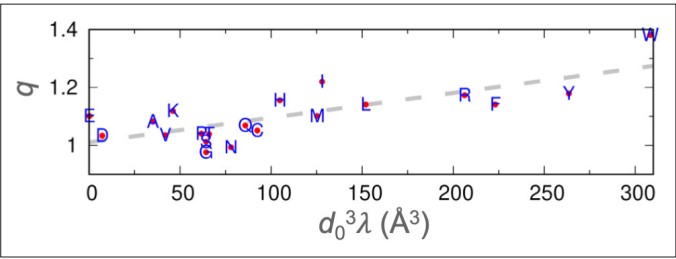

**Figure 9.** Correlation between the stickiness parameters ($\lambda$) and the NMR relaxation parameters ($q$). The regression line is shown as dashes.

The online version of this article includes the following source data for figure 9:

**Source data 1.** Source data for *Figure 9*.

**Table 2.** RMSEs ($s^{-1}$) of $R_2$ predictions by SeqDYN and MD for 10 IDPs.

| IDP name | SeqDYN | MD |
|---|---|---|
| A1-LCD | 0.60* | 0.59 [§, ¶] |
| Aβ40 | 0.38* | 0.38 [§] |
| HOX-DFD | 1.99* | 1.40 [§] |
| α-synuclein | 0.44* | 0.50 [§] |
| p53TAD | 0.33* | 1.04 [**] |
| Pup | 0.43* | 1.00** |
| Sev-NT | 0.38*,[†] | 1.10 [§,††] |
| tau K18 | 0.83* | 0.80 [§] |
| ChiZ | 0.74 [‡] | 1.40 [‡‡] |
| FtsQ | 1.71 [§,†] | 1.70 [§§] |

*Based on leave-one-out training (using 44 IDPs).

[†]Helix boost applied.

[‡]Based on training by the full training set (45 IDPs).

[§]From **Dey et al., 2022**.

[¶]RMSE is scaled down by a factor of 2.39, to correct for the effect of temperature (MD at 288 K; see **Figure 3—figure supplement 1C**).

[**]From **Yu and Brüschweiler, 2022**.

[††]RMSE is scaled down by a factor of 2.99, to correct for the effects of temperature and magnetic field (MD at 274 K and 850 MHz; see **Figure 3—figure supplement 1B**).

[‡‡]Originally calculated in **Hicks et al., 2020** with correction in **Hicks et al., 2021**.

[§§]From **Smrt et al., 2023**.

a qualitative indicator of $R_2$. Here again the calculation was based on an additive model. **Sekiyama et al., 2022** employed a multiplicative model, with $R_2$ calculated as a geometric mean of 'indices of local dynamics' over a five-residue window. These indices, akin to our $q$ parameters, were trained on a single IDR (TIA-1 prion-like domain) and used to reproduce the measured $R_2$ for the same IDR. As we have illustrated on CBP-ID4 (**Figure 5—figure supplement 1**), training on a single protein merely biases the parameters to that model and has little value in predicting $R_2$ for other proteins. In comparison, SeqDYN is trained on 45 IDPs and its predictions are robust and achieve quantitative agreement with measured $R_2$.

Ten of the IDPs tested here have been studied recently by MD simulations using IDP-specific force fields (**Dey et al., 2022**; **Hicks et al., 2020**; **Yu and Brüschweiler, 2022**; **Smrt et al., 2023**). In **Table 2**, we compare the RMSEs of SeqDYN predictions with those for $R_2$ calculations from MD simulations. For five of these IDPs: A1-LCD, Aβ40, α-synuclein, tau K18, and FtsQ, RMSEs of SeqDYN and MD are remarkably similar. Four of these IDPs lack significant population of α-helices or β-sheets, but FtsQ forms a stable long helix. For one other IDP, namely HOX-DFD, MD, by explicitly modeling its folded domain, does a much better job in predicting $R_2$ than SeqDYN (RMSEs of 1.40 $s^{-1}$ vs 1.99 $s^{-1}$). However, for the four remaining IDPs: p53TAD, Pup, Sev-NT, and ChiZ, SeqDYN significantly outperforms MD, with RMSEs averaging only 0.47 $s^{-1}$, compared to the MD counterpart of 1.14 $s^{-1}$. Overall, SeqDYN is very competitive against MD in predicting $R_2$, but without the significant computational cost. While MD simulations can reveal details of local interactions, as noted for α-synuclein, and capture tertiary interactions if they occur, they still suffer from perennial problems of force-field imperfection and inadequate sampling. SeqDYN provides an accurate description of IDP dynamics at a 'mean-field' level, but could miss idiosyncratic behaviors of specific local sequences.

Deep-learning models have become very powerful, but they usually have millions of parameters and require millions of protein sequences for training (**Rives et al., 2021**). In contrast, SeqDYN employs a mathematical model with dozens of parameters and requires only dozens of proteins for training. Reduced models (by collapsing amino acids into a small number of distinct types) have even been trained on <10 IDPs to predict propensities for binding nanoparticles (**Li et al., 2020**) or membranes

(*Qin et al., 2022*). The mathematical model-based approach may be useful in other applications where data, similar to $R_2$, are limited, including predictions of IDP secondary chemical shifts or residues that bind drug molecules (*Robustelli et al., 2022*) or protein targets, or even in protein design, for example for recognizing an antigenic site or a specific DNA site.

## Methods
### Collection of IDPs with measured $R_2$

Starting from six nonhomologous IDPs in our previous MD study (*Dey et al., 2022*), we obtained $R_2$ data for eight IDPs from the Bimolecular Magnetic Resonance Data Bank (BMRB; https://bmrb.io); data for two other IDPs were from our collaborators (*Hicks et al., 2020*; *Smrt et al., 2023*). Most of the 54 IDPs studied here were from searching the literature. Disorder was judged by dispersion in backbone amide proton chemical shifts, NOE, and SCS. $R_2$ data that were not available from the authors or BMRB were obtained by digitizing $R_2$ plots presented in figures of published papers, using WebPlotDigitizer (https://automeris.io/WebPlotDigitizer; *Rohatgi, 2022*) and further inspected visually.

Homology of IDPs was checked by sequence alignment using Clustal W (http://www.clustal.org/clustal2; *Larkin et al., 2007*), and presented as a clock-like tree using the 'ape' package (http://ape-package.ird.fr; *Paradis et al., 2004*). IDPs that had discernible homology with the selected training set were removed. Removed IDPs included HOX-SCR and β-synuclein from our previous MD study (*Dey et al., 2022*), due to homology with HOX-DFD and α-synuclein, respectively.

### Coding for SeqDYN

The training of SeqDYN was coded in python, similar to our previous work for predicting residue-specific membrane association propensities (ReSMAP; https://zhougroup-uic.github.io/ReSMAPidp/; *Qin et al., 2022*). The cost function was the sum of mean-squared-errors for the IDPs in the training set. We used the least_squares function in scipy.optimize, with Trust Region Reflective as the minimization algorithm and all parameters restricted to the positive range. For the web server (https://zhougroup-uic.github.io/SeqDYNidp/; *Qin and Zhou, 2024*), we rewrote the prediction code javascript.

## Acknowledgements

This work was supported by Grant GM118091 from the National Institutes of Health.

## Additional information

### Funding

| Funder | Grant reference number | Author |
| --- | --- | --- |
| National Institutes of Health | GM118091 | Huan-Xiang Zhou |

The funders had no role in study design, data collection and interpretation, or the decision to submit the work for publication.

### Author contributions

Sanbo Qin, Resources, Data curation, Software, Formal analysis, Validation, Investigation, Visualization, Methodology; Huan-Xiang Zhou, Conceptualization, Resources, Data curation, Formal analysis, Supervision, Funding acquisition, Validation, Investigation, Visualization, Methodology, Writing - original draft, Project administration, Writing - review and editing

### Author ORCIDs

Huan-Xiang Zhou (iD) https://orcid.org/0000-0001-9020-0302

Reviewer #2 (Public review): https://doi.org/10.7554/eLife.88958.3.sa1

Reviewer #3 (Public review): https://doi.org/10.7554/eLife.88958.3.sa2
Author response https://doi.org/10.7554/eLife.88958.3.sa3

## Additional files

### Supplementary files
• MDAR checklist

### Data availability
All data generated or analyzed during this study are included in the manuscript and supplementary files; source data have been provided for *Figures 3–9*, *Figure 3—figure supplement 1*, *Figure 4—figure supplement 1*, and *Figure 5—figure supplement 1*.

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

## Appendix 1

### Sequences of 54 IDPs
Terminal tags and other insertions are underlined.

Training set (45 IDPs)

>A1-LCD (Uniprot ID P04256, residues 186-320; deletion of 258-263)
GSMASASSSQRGRSGSGNFGGGRGGGFGGNDNFGRGGNFSGRGGFGGSRGGGGYGGSGDGYNGFGNDGSNFG
GGGNYNNQSSNFGPMKGGNFGGRSSGPYGGGGQYFAKPRNQGGYGGSSSSSSYGSGRRF
>Aβ40 (Uniprot ID Q28053, residues 7-46)
DAEFRHDSGYEVHHQKLVFFAEDVGSNKGAIIGLMVGGV
>Ash1 (Uniprot ID P34233, residues 420-500)
GASASSSPSPSTPTKSGKMRSRSSSPVRPKAYTPSPRSPNYHRFALDSPPQSPRRSSNSSITKKGSRRSSGS
SPTRHTTRVCV
>Beclin1 (Uniprot ID Q14457, residues 1-150)
MGSSHHHHHHSQDPMEGSKTSNNSTMQVSFVSQRSSQPLKLDTSFKILDRVTIQELTAPLLTTAQAKPGETQ
EEETNSGEEPFIETPRQDGVSRRFIPPARMMSTESANSFTLIGEASDGGTMENLSRRLKVTGDLFDIMSGQT
DVDHPLSEESTDTLLDQLDTY
>CAPRIN1 (Uniprot ID Q14444, residues 607-709)
SRGVSRGGSRGARGLMNGYRGPANGFRGGYDGYRPSFSNTPNSGYTQSQFSAPRDYSGYQRDGYQQNFKRGS
GQSGPRGAPRGRGGPPRPNRGMPQMNTQQVN
>CBP-ID4 (Uniprot ID Q92793, residues 1852-2057)
MQQQIQHRLQQAQLMRRRMATMNTRNVPQQSLPSPTSAPPGTPTQQPSTPQTPQPPAQPQPSPVSMSPAGFP
SVARTQPPTTVSTGKPTSQVPAPPPPAQPPPAAVEAARQIEREAQQQQHLYRVNINNSMPPGRTGMGTPGSQ
MAPVSLNVPRPNQVSGPVMPSMPPGQWQQAPLPQQQPMPGLPRPVISMQAQAAVAGPRMPSVQ
>GbnD4-DHD (Uniprot ID A0A808VWJ6, residues 412-482)
MKHHHHHHHHGGLVPRGSHGSDEGVPDALRADTVPRAGPVRYARRRYWIGEARSDALAPAAPLEREPLPAEA
MGAYFAIRRTDADDTVAAH
>ERD14 (Uniprot ID P42763, residues 1-185)
MAEEIKNVPEQEVPKVATEESSAEVTDRGLFDFLGKKKDETKPEETPIASEFEQKVHISEPEPEVKHESLLE
KLHRSDSSSSSSSEEEGSDGEKRKKKKEKKKPTTEVEVKEEEKKGFMEKLKEKLPGHKKPEDGSAVAAAPVV
VPPPVEEAHPVEKKGILEKIKEKLPGYHPKTTVEEEKKDKE
>ExsE (Uniprot ID Q9I322, residues 1-81)
MKIESIPPVQPSQDAGAEAVGHFEGRSVTRAAVRGDDRSSVAGLARWLARNVAGDPRSEQALQRLADGDGTP
LEARTVRRREFLEGSS
>FCP1 (Uniprot ID Q9Y5B0, residues 879-961)
PGPEEQEEEPQPRKPGTRRERTLGAPASSERSAAGGRGPRGHKRKLNEEDAASESSRESSNEDEGSSSEADE
MAKALEAELNDLM
>FUS (Uniprot ID P35637, residues 1-163)
MASNDYTQQATQSYGAYPTQPGQGYSQQSSQPYGQQSYSGYSQSTDTSGYGQSSYSSYGQSQNTGYGTQSTP
QGYGSTGGYGSSQSSQSSYGQQSSYPGYGQQPAPSSTSGSYGSSSQSSSYGQPQSGSYSQQPSYGGQQQSYG
QQQSYNPPQGYGQQNQYNS
>GAb1 (Uniprot ID B7Z3B9, residues 510-591)
SSPMIKPKGDKQVEYLDLDLDSGKSTPPRKQKSSGSGSSVADERVDYVVVDQQKTLALKSTREAWTDGRQST
ESETPAKSVK
>hACTR (Uniprot ID Q9Y6Q9, residues 1023-1091)
GTQNRPLLRNSLDDLVGPPSNLEGQSDERALLDQLHTLLSNTDATGLEEIDRALGIPELVNQGQ
ALEPK>Hahellin (Uniprot ID Q2SHN6, residues 162-252)
MGEKTVKLYEDTHFKGYSVELPVGDYNLSSLISRGALNDDLSSARVPSGLRLEVFQHNNFKGVRDFYTSDAA
ELSRDNDASSVRVSKMETTN
>hCSD1 (Uniprot ID P20810, residues 137-277)
AVPVESKPDKPSGKSGMDAALDDLIDTLGGPEETEEENTTYTGPEVSDPMSSTYIEELGKREVTIPPKYREL
LAKKEGITGPPADSSKPIGPDDAIDALSSDFTCGSPTAAGKKTEKEESTEVLKAQSAGTVRSAAPPQEK
>HOX-DFD (Uniprot ID P07548, residues 337-426)

```
TDGERIIYPWMKKIHVAGVANGSYQPGMEPKRQRTAYTRHQILELEKEFHYNRYLTRRRRIEIAHTLVLSER
QIKIWFQNRRMKWKKDNK
```
>hZIP4-ICL2 (Uniprot ID Q6P5W5, residues 424-498)
```
GDRGPEFELGTLPRDPEDLEDGPCGHSSHSHGGHSHGVSLQLAPSELRQPKPPHEGSRADLVAEESPELLNP
EPRRLSPELRLLPYGHGLSAWSHPQFEK
```
>Jaburetox (Uniprot ID I1K3K3, residues 230-320)
```
MGPVNEANCKAAMEIVCRREFGHKEEEDASEGVTTGDPDCPFTKAIPREEYANKYGPTIGDKIRLGDTDLIA
EIEKDFALYGDESVFGGGKVIH
```
>KRS-NT (Uniprot ID Q15046, residues 1-72)
```
MAAVQAAEVKVDGSEPKLSKNELKRRLKAEKKVAEKEAKQKELSEKQLSQATAAATNHTTDNGVGPEEESVD
```
>MBP-xα2 (Uniprot ID P04370, residues 172-237)
```
SIGRFFSGDRGAPKRGSGKDSHTRTTHYGSLPQKSQHGRTQDENPVVHFFKNIVTPRTPPPSQGKGRGLS
```
>MKK4 (Uniprot ID P45985, residues 1-86)
```
MAAPSPSGGGGSGGGSGSGTPGPVGSPAPGHPAVSSMQGKRKALKLNFANPPFKSTARFTLNPNPTGVQNPH
IERLRTHSIESSGK
```
>N-Cby (Uniprot ID B0QY54, residues 1-63)
```
MPFFGNTFSPKKTPPRKSASLSNLHSLDRSTREVELGLEYGSPTMNLAGQSLKFENGQWIAET
```
>Niv-PNTD (Uniprot ID P0C1C7, residues 1-406)
```
MDKLELVNDGLNIIDFIQKNQKEIQKTYGRSSIQQPSIKDQTKAWEDFLQCTSGESEQVEGGMSKDDGDVER
RNLEDLSSTSPTDGTIGKRVSNTRDWAEGSDDIQLDPVVTDVVYHDHGGECTGYGFTSSPERGWSDYTSGAN
NGNVCLVSDAKMLSYAPEIAVSKEDRETDLVHLENKLSTTGLNPTAVPFTLRNLSDPAKDSPVIAEHYYGLG
VKEQNVGPQTSRNVNLDSIKLYTSDDEEADQLEFEDEFAGSSSEVIVGISPEDEEPSSVGGKPNESIGRTIE
GQSIRDNLQAKDNKSTDVPGAGPKDSAVKEEPPQKRLPMLAEEFECSGSEDPIIRELLKENSLINCQQGKDA
QPPYHWSIERSISPDKTEIVNGAVQTADRQRPGTPMPKSRGIPIKK
```
>NS5A-D2D3 (Uniprot ID O92972, residues 2163-2419)
```
GHMASGSLRGGEPEPDVTVLTSMLTDPSHITAETAKRRLARGSPPSLASSSASQLSAPSLKATCTTHHDSPD
ADLIEANLLWRQEMGGNITRVESENKVVILDSFEPLHADGDEREISVAAEILRKSRKFPSALPIWARPDYNP
PLLESWKDPDYVPPVVHGCPLPPTKAPPIPPPRRKRTVVLTESNVSSALAELATKTFGSSGSSAVDSGTATA
LPDQASDDGDKGSDVESYSSMPPLEGEPGDPDLSDGSWSTVSEEASEDVVCC
```
>NUPR1 (Uniprot ID O60356, residues 2-82)
```
MRGSHHHHHHGSATFPPATSAPQQPPGPEDEDSSLDESDLYSLAHSYLGGGGRKGRTKREAAANTNRPSPGG
HERKLVTKLQNSERKKRGARR
```
OPN (Uniprot ID F1NSM8, residues 46-264)
```
MHQDHVDSQSQEHLQQTQNDLASLQQTHYSSEENADVPEQPDFPDVPSKSQETVDDDDDDDNDSNDTDESDE
VFTDFPTEAPVAPFNRGDNAGRGDSVAYGFRAKAHVVKASKIRKAARKLIEDDATTEDGDSQPAGLWWPKES
REQNSRELPQHQSVENDSRPKFDSREVDGGDSKASAGVDSRESQGSVPAVDASNQTLESAEDAEDRHS
IENNEVTR
```
>p53TAD (Uniprot ID P04637, residues 1-71)
```
MEEPQSDPSVEPPLSQETFSDLWKLLPENNVLSPLPSQAMDDLMLSPDDIEQWFTEDPGPDEAPRMPEAAPRV
```
>PDEγ (Uniprot ID P61248, residues 1-87)
```
MNLEPPKAEIRSATRVMGGPVTPRKGPPKFKQRQTRQFKSKPPKKGVQGFGDDIPGMEGLGTDITVIAPWEA
FNHLELHELAQYGII
```
>PKIα (Uniprot ID P61925, residues 2-76)
```
TDVETTYADFIASGRTGRRNAIHDILVSSASGNSNELALKLAGLDINKTEGEEDAQRSSTEQSGEAQGEAAKSES
```
>Mev-PNTD (Uniprot ID P03422, residues 1-304)
```
MAEEQARHVKNGLECIRALKAEPIGSLAIEEAMAAWSEISDNPGQERATCREEKAGSSGLSKPCLSAIGSTE
GGAPRIRGQGPGESDDDAETLGIPPRNLQASSTGLQCYYVYDHSGEAVKGIQDADSIMVQSGLDGDSTLSGG
DNESENSDVDIGEPDTEGYAITDRGSAPISMGFRASDVETAEGGEIHELLRLQSRGNNFPKLGKTLNVPPPP
DPGRASTSGTPIKKGTERRLASFGTEIASLLTGGATQCARKSPSEPSGPGAPAGNVPECVSNAALIQEWTPE
SGTTISPRSQNNEEGG
```
>ProTα (Uniprot ID P06454, residues 1-111)
```
GPMSDAAVDTSSEITTKDLKEKKEVVEEAENGRDAPANGNAENEENGEQEADNEVDEEEEEGGEEEEEEEEG
DGEEEDGDEDEEAESATGKRAAEDDEDDDVDTKKQKTDEDD
```
>Pup (Uniprot ID P9WHN4, residues 1-64)
```
MAQEQTKRGGGGGGDDDDIAGSTAAGQERREKLTEETDDLLDEIDDVLEENAEDFVRAYVQKGGQ
```
>rmBG21 (Uniprot ID P04370, residues 2-190)
```

GNHSGKRELSAEKASKDGEIHRGEAGKKRSVGKLSQTASEDSDVFGEADAIQNNGTSAEDTAVTDSKHTADP
KNNWQGAHPADPGNRPHLIRLFSRDAPGREDNTFKDRPSESDELQTIQEDPTAASGGLDVMASQKRPSQRSK
YLATASTMDHARHGFLPRHRDTGILDSIGRFFSGDRGAPKRGSGKVSLEHHHHHH
>RPB1 (Uniprot ID P24928, residues 1773-1970)
GHMSPNYTPTSPNYSPTSPSYSPTSPSYSPTSPSYSPSSPRYTPQSPTYTPSSPSYSPSSPSYSPASPKYTP
TSPSYSPSSPEYTPTSPKYSPTSPKYSPTSPKYSPTSPTYSPTTPKYSPTSPTYSPTSPVYTPTSPKYSPTS
PTYSPTSPKYSPTSPTYSPTSPKGSTYSPTSPGYSPTSPTYSLTSPAISPDDSDEEN
>securin (Uniprot ID O95997, residues 1-202)
MATLIYVDKENGEPGTRVVAKDGLKLGSGPSIKALDGRSQVSTPRFGKTFDAPPALPKATRKALGTVNRATE
KSVKTKGPLKQKQPSFSAKKMTEKTVKAKSSVPASDDAYPEIEKFFPFNPLDFESFDLPEEHQIAHLPLSGV
PLMILDEERELEKLFQLGPPSPVKMPSPPWESNLLQSPSSILSTLDVELPPVCCDIDI
>Sev-NT (Uniprot ID Q07097, residues 401-524)
LSGGDGAYHEPTGGGAIEVALDNADIDLETEAHADQDARGWGGESGERWARQVSGGHFVTLHGAERLEEETN
DEDVSDIERRIAMRLAERRQEDSATHGDEGRNNGVDHDEDDDAAAVAGIGGI
>Sic1 (Uniprot ID P38634, residues 1-90)
GSMTPSTPPRSRGTRYLAQPSGNTSSSALMQGQKTPQKPSQNLVPVTPSTTKSFKNAPLLAPPNSNMGMTSP
FNGLTSPQRSPFPKSSVKRT
>SKIPN (Uniprot ID G3V5R3, residues 59-129)
GDGGAFPEIHVAQYPLDMGRKKKMSNALAIQVDSEGKIKYDAIARQGQSKDKVIYSKYTDLVPKEVMNADD
>SLBP-NT (Uniprot ID Q9VAN6, residues 17-108)
MGSSHHHHHHSSGLVPRGSHMGSGSLNSSASSISIDVKPTMQSWAQEVRAEFGHSDEASSSLNSSAASCGSL
AKKETADGNLESKDGEGREMAFEFLDGVNEVKFERLVKEEK
>α-synuclein (Uniprot ID P37840, residues 1-140)
MDVFMKGLSKAKEGVVAAAEKTKQGVAEAAGKTKEGVLYVGSKTKEGVVHGVATVAEKTKEQVTNVGGAVVT
GVTAVAQKTVEGAGSIAAATGFVKKDQLGKNEEGAPQEGILEDMPVDPDNEAYEMPSEEGYQDYEPEA
>SOCS5-JIR (Uniprot ID A0A5E4BAI0, residues 12-81)
RSLRQRLQDTVGLCFPMRTYSKQSKPLFSNKRKIHLSELMLEKCPFPAGSDLAQKWHLIKQHTAPVSPHS
>tau K18 (Uniprot ID Q9MYX8, residues 186-314)
QTAPVPMPDLKNVKSKIGSTENLKHQPGGGKVQIINKKLDLSNVQSKCGSKDNIKHVPGGGSVQIVYKPVDL
SKVTSKAGSLGNIHHKPGGGQVEVKSEKLDFKDRVQSKIGSLDNITHVPGGGNKKIE
>TC1 (Uniprot ID Q9NR00, residues 1-106)
MKAKRSHQAIIMSTSLRVSPSIHGYHFDTASRKKAVGNIFENTDQESLERLFRNSGDKKAEERAKIIFAIDQ
DVEEKTRALMALKKRTKDKLFQFLKLRKYSIKVH
>TDP-43 (Uniprot ID Q13148, residues 267-414)
GHMNRQLERSGRFGGNPGGFGNQGGFGNSRGGGAGLGNNQGSNMGGGMNFGAFSINPAMMAAAQAALQSSWG
MMGMLASQQNQSGPSGNNQNQGNMQREPNQAFGSGNNSYSGSNSGAAIGWGSASNAGSGSGFNGGFGSSMDS
KSSGWGM
>γ-tubulin-CT (Uniprot ID P53378, residues 439-473)
LLRGAAEQDSYLDDVLVDDENMVGELEEDLDADGDHKLV

## Test set (9 IDPs)

>AMOTL1 (Uniprot ID Q8IY63, residues 178-384)
STQPQQNNEELPTYEEAKAQSQFFRGQQQQQQQQGAVGHGYYMAGGTSQKSRTEGRPTVNRANSGQAHKDEA
LKELKQGHVRSLSERIMQLSLERNGAKQHLPGSGNGKGFKVGGGPSPAQPAGKVLDPRGPPPEYPFKTKQMM
SPVSKTQEHGLFYGDQHPGMLHEMVKPYPAPQPVRTDVAVLRYQPPPEYGVTSRPCQLPFPST
>CAHS-8 (Uniprot ID P0CU50, residues 1-227)
MSGRNVESHMERNEKVVVNNSGHADVKKQQQQVEHTEFTHTEVKAPLIHPAPPIISTGAAGLAEEIVGQGFT
ASAARISGGTAEVHLQPSAAMTEEARRDQERYRQEQESIAKQQEREMEKKTEAYRKTAEAEAEKIRKELEKQ
HARDVEFRKDLIESTIDRQKREVDLEAKMAKRELDREGQLAKEALERSRLATNVEVNFDSAAGHTVSGGTTV
STSDKMEIKRNENLYFQ
>ChiZ (Uniprot ID I6YA32, residues 1-64)
MTPVRPPHTPDPLNLRGPLDGPRWRRAEPAQSRRPGRSRPGGAPLRYHRTGVGMSRTGHGSRPV
>-endosulfine (Uniprot ID O43768, residues 1-121)

MSQKQEEENPAEETGEEKQDTQEKEGILPERAEEAKLKAKYPSLGQKPGGSDFLMKRLQKGQKYFDSGDYNM
AKAKMKNKQLPSAGPDKNLVTGDHIPTPQDLPQRKSSLVTSKLAGGQVE
>FtsQ (Uniprot ID Q8IY63, residues 1-99)
MTEHNEDPQIERVADDAADEEAVTEPLATESKDEPAEHPEFEGPRRRARRERAERRAAQARATAIEQARRAA
KRRARGQIVSEQNPAKPAARGVVRGLK
>Pdx1 (Uniprot ID P52945, residues 204-283)
GPGEEDKKRGGGTAVGGGGVAEPEQDCAVTSGEELLALPPPPPPGGAVPPAAPVAAREGRLPPGLSASPQPS
SVAPRRPQEPR
>synaptobrevin-2 (Uniprot ID P63027, residues 1-96)
MSATAATAPPAAPAGEGGPPAPPPNLTSNRRLQQTQAQVDEVVDIMRVNVDKVLERDQKLSELDDRADALQA
GASQFETSAAKLKRKYWWKNLKMM
>TIA-1 (Uniprot ID P31483, residues 320-386)
MGSSHHHHHHHHHHHHHSENLYFQGGQYVPNGWQVPAYGVYGQPWSQQGFNQTQSSAPWMGPNYSVPPPQGQN
GSMLPSQPAGYRVAGYETQ
>YAP (Uniprot ID P46937, residues 50-171)
AGHQIVHVRGDSETDLEALFNAVMNPKTANVPQTVPMRLRKLPDSFFKPPEPKSHSRQASTDAGTAGALTPQ
HVRAHSSPASLQLGAVSPGTLTPTGVVSGPAATPTAQHLRQSSFEIPDDV

