## [Editor Report · eLife assessment]

In this **useful** study, a **solid** machine learning approach based on a broad set of systems to predict the R2 relaxation rates of residues in intrinsically disordered proteins (IDPs) is described. The ability to predict the patterns of R2 will be helpful to guide experimental studies of IDPs. A potential weakness is that the predicted R2 values may include both fast and slow motions, thus the predictions provide only limited new physical insights into the nature of the underlying protein dynamics, such as the most relevant timescale.

---

## [Referee Report · Reviewer #2 (Public review)]

Qin, Sanbo and Zhou, Huan-Xiang created a model, SeqDYN, to predict nuclear magnetic resonance (NMR) spin relaxation spectra of intrinsically disordered proteins (IDPs), based primarily on amino acid sequence. To fit NMR data, SeqDYN uses 21 parameters, 20 that correspond to each amino acid, and a sequence correlation length for interactions. The model demonstrates that local sequence features impact the dynamics of the IDP, as SeqDYN performs better than a one residue predictor, despite having similar numbers of parameters. SeqDYN is trained using 45 IDP sequences and is retrained using both leave-one-out cross validation and five-fold cross validation, ensuring the model's robustness. While SeqDYN can provide reasonably accurate predictions in many cases, the authors note that improvements can be made by incorporating secondary structure predictions, especially for alpha-helices that exceed the correlation length of the model. The authors apply SeqDYN to study nine IDPs and a denatured ordered protein, demonstrating its predictive power. The model can be easily accessed via the website mentioned in the text.

The authors have adequately addressed the majority of my previous concerns. However, I still wonder if an attempt to fit the individual protein fitting parameter based on temperature and magnetic field strength would be possible. The authors would have 45 data points on which to fit such a parameter, which would only depend on two variables.

---

## [Referee Report · Reviewer #3 (Public review)]

The revised manuscript adds some new relevant analyses. It still, however, is unclear which timescales of motions the method refers to and there is confusion about whether the model can predict "slower motions". While the authors answer some of my points, others are left unanswered. That is of course the authors' prerogative, and readers will in any case be able to read the reviewer comments. I am not sure it is productive to add further comments at this point.

Below are my comments from the first round of review:

The manuscript by Qin and Zhou presents an approach to predict dynamical properties of an intrinsically disordered protein (IDP) from sequence alone. In particular, the authors train a simple (but useful) machine learning model to predict (rescaled) NMR R2 values from sequence. Although these R2 rates only probe some aspects of IDR dynamics and the method does not provide insight into the molecular aspects of processes that lead to perturbed dynamics, the method can be useful to guide experiments.

A strength of the work is that the authors train their model on an observable that directly relates to protein dynamics. They also analyse a relatively broad set of proteins which means that one can see actual variation in accuracy across the proteins.

A weakness of the work is that it is not always clear what the measured R2 rates mean. In some cases, these may include both fast and slow motions (intrinsic R2 rates and exchange contributions). This in turn means that it is actually not clear what the authors are predicting. The work would also be strengthened by making the code available (in addition to the webservice), and by making it easier to compare the accuracy on the training and testing data.

---

## [Author Response]

The following is the authors’ response to the original reviews.

In this useful study, a solid machine learning approach based on a broad set of systems to predict the R2 relaxation rates of residues in intrinsically disordered proteins (IDPs) is described. The ability to predict the patterns of R2 will be helpful to guide experimental studies of IDPs. A potential weakness is that the predicted R2 values may include both fast and slow motions, thus the predictions provide only limited new physical insights into the nature of the relevant protein dynamics.

Fast motions are less sequence-dependent (e.g., as shown by R1). Hence the sequence-dependent part of R2 singles out slow motion.

**Public Reviews:**

**Reviewer #1 (Public Review):**
Solution state 15N backbone NMR relaxation from proteins reports on the reorientational properties of the N-H bonds distributed throughout the peptide chain. This information is crucial to understanding the motions of intrinsically disordered proteins and as such has focussed the attention of many researchers over the last 20-30 years, both experimentally, analytically and using numerical simulation.This manuscript proposes an empirical approach to the prediction of transverse 15N relaxation rates, using a simple formula that is parameterised against a set of 45 proteins. Relaxation rates measured under a wide range of experimental conditions are combined to optimize residuespecific parameters such that they reproduce the overall shape of the relaxation profile. The purely empirical study essentially ignores NMR relaxation theory, which is unfortunate, because it is likely that more insight could have been derived if theoretical aspects had been considered at any level of detail.

NMR relaxation theory is very valuable in particular regarding motions on different timescales. However, it has very little to say about the sequence dependence of slow motions, which is the focus of our work.

Despite some novel aspects, in particular the diversity of the relaxation data sets, the residuespecific parameters do not provide much new insight beyond earlier work that has also noted that sidechain bulkiness correlated with the profile of R2 in disordered proteins.

The novel insight from our work is that R2 can mostly be predicted based on the local sequence.

Nevertheless, the manuscript provides an interesting statistical analysis of a diverse set of deposited transverse relaxation rates that could be useful to the community.

Thank you!

Crucially, and somewhat in contradiction to the authors stated aims in the introduction, I do not feel that the article delivers real insight into the nature of IDP dynamics. Related to this, I have difficulty understanding how an approximate prediction of the overall trend of expected transverse relaxation rates will be of further use to scientists working on IDPs. We already know where the secondary structural elements are (from 13C chemical shifts which are essential for backbone assignment) and the necessary 'scaling' of the profile to match experimental data actually contains a lot of the information that researchers seek.

Again, the novel insight is that slow motions that dictate the sequence dependence of R2 can mostly be predicted based on the local sequence. The scaling factor may contain useful information but does not tell us anything about the sequence dependence of IDP dynamics.

This reviewer brings up a lot of valuable points, clearly from an NMR spectroscopist’s perspective. The emphasis of our paper is somewhat different from that perspective. For example, we were interested in whether tertiary contacts make significant contributions to R2, as sometimes claimed. Our results show that, in general, they do not; instead local contacts dominate the sequence dependence of R2.

(1) The introduction is confusing, mixing different contributions to R2 as if they emanated from the same physics, which is not necessarily true. 15N transverse relaxation is said to report on 'slower' dynamics from 10s of nanoseconds up to 1 microsecond. Semi-classical Redfield theory shows that transverse relaxation is sensitive to both adiabatic and non-adiabatic terms, due to spin state transitions induced by stochastic motions, and dephasing of coherence due to local field changes, again induced by stochastic motions. These are faster than the relaxation limit dictated by the angular correlation function. Beyond this, exchange effects can also contribute to measured R2. The extent and timescale limit of this contribution depends on the particular pulse sequence used to measure the relaxation. The differences in the pulse sequences used could be presented, and the implications of these differences for the accuracy of the predictive algorithm discussed.

Indeed pulse sequences affect the measured R2 values. We make the modest assumption that such experimental idiosyncrasy would not corrupt the sequence dependence of IDP dynamics. As for exchange effects, our expectation is that the current SeqDYN may not do well for R2s where slow exchange plays a dominant role in generating sequence dependence, as tertiary contacts would be prominent in those cases; we now present one such case (new Fig. S5).

(2) Previous authors have noted the correlation between observed transverse relaxation rates and amino acid sidechain bulkiness. Apart from repeating this observation and optimizing an apparently bulkiness-related parameter on the basis of R2 profiles, I am not clear what more we learn, or what can be derived from such an analysis. If one can possibly identify a motif of secondary structure because raised R2 values in a helix, for example, are missed from the prediction, surely the authors would know about the helix anyway, because they will have assigned the 13C backbone resonances, from which helical propensity can be readily calculated.

We think that a sequence-based method that is demonstrated to predict well R2 values from expensive NMR experiments is significant. That pi-pi and cation-pi interactions are prominent features of local contacts and may seed tertiary contacts and mediate inter-chain contacts that drive phase separation is a valuable insight.

(3) Transverse relaxation rates in IDPs are often measured to a precision of 0.1s-1 or less. This level of precision is achieved because the line-shapes of the resonances are very narrow and high resolution and sensitivity are commonly measurable. The predictions of relaxation rates, even when applying uniform scaling to optimize best-agreement, is often different to experimental measurement by 10 or 20 times the measured accuracy. There are no experimental errors in the figures. These are essential and should be shown for ease of comparison between experiment and prediction.

Again, our focus is not the precision of the absolute R2 values, but rather the sequence dependence of R2.

(4) The impact of structured elements on the dynamic properties of IDPs tethered to them is very well studied in the literature. Slower motions are also increased when, for example the unfolded domain binds a partner, because of the increased slow correlation time. The ad hoc 'helical boosting' proposed by the authors seems to have the opposite effect. When the helical rates are higher, the other rates are significantly reduced. I guess that this is simply a scaling problem. This highlights the limitation of scaling the rates in the secondary structural element by the same value as the rest of the protein, because the timescales of the motion are very different in these regions. In fact the scaling applied by the authors contains very important information. It is also not correct to compare the RMSD of the proposed method with MD, when MD has not applied a 'scaling'. This scaling contains all the information about relative importance of different components to the motion and their timescales, and here it is simply applied and not further analysed.

Actually, applying the boost factor achieves the effect of a different scaling factor for the secondary structure element than for the rest of the protein.

Regarding comparing RMSEs of SeqDYN and MD, it is true that SeqDYN applies a scaling factor whereas MD does not. However, even if we apply scaling to MD results it will not change the basic conclusion that “SeqDYN is very competitive against MD in predicting _R_2, but without the significant computational cost.”

(5) Generally, the uniform scaling of all values by the same number is serious oversimplification. Motions are happening on all timescales they are giving rise to different transverse relaxation. It is not possible to describe IDP relaxation in terms of one single motion. Detailed studies over more than 30 years, have demonstrated that more than one component to the autocorrelation function is essential in order to account for motions on different timescales in denatured, partially disordered or intrinsically unfolded states. If one could 'scale' everything by the same number, this would imply that only one timescale of motion were important and that all others could be neglected, and this at every site in the protein. This is not expected to be the case, and in fact in the examples shown by the authors it is also never the case. There are always regions where the predicted rates are very different from experiment (with respect to experimental error), presumably because local dynamics are occurring on different timescales to the majority of the molecule. These observations contain useful information, and the observation that a single scaling works quite well probably tells us that one component of the motion is dominant, but not universally. This could be discussed.

The reviewer appears to equate a single scaling factor with a single type of motion -- this is not correct. A single scaling factor just means that we factor out effects (e.g., temperature or magnetic field) that are uniform across the IDP sequence.

(6) With respect to the accuracy of the prediction, discussion about molecular detail such as pi-pi interactions and phase separation propensity is possibly a little speculative.

It is speculative; we now add more support to this speculation (p. 18 and new Fig. S6).

(7) The authors often declare that the prediction reproduces the experimental data. The comparisons with experimental data need to be presented in terms of the chi2 per residue, using the experimentally measured precision which as mentioned, is often very high.

Again, our interest is the sequence dependence of R2, not the absolute R2 value and its measurement precision.

**Reviewer #2 (Public Review):**
Qin, Sanbo and Zhou, Huan-Xiang created a model, SeqDYN, to predict nuclear magnetic resonance (NMR) spin relaxation spectra of intrinsically disordered proteins (IDPs), based primarily on amino acid sequence. To fit NMR data, SeqDYN uses 21 parameters, 20 that correspond to each amino acid, and a sequence correlation length for interactions. The model demonstrates that local sequence features impact the dynamics of the IDP, as SeqDYN performs better than a one residue predictor, despite having similar numbers of parameters. SeqDYN is trained using 45 IDP sequences and is retrained using both leave-one-out cross validation and five-fold cross validation, ensuring the model's robustness. While SeqDYN can provide reasonably accurate predictions in many cases, the authors note that improvements can be made by incorporating secondary structure predictions, especially for alpha-helices that exceed the correlation length of the model. The authors apply SeqDYN to study nine IDPs and a denatured ordered protein, demonstrating its predictive power. The model can be easily accessed via the website mentioned in the text.While the conclusions of the paper are primarily supported by the data, there are some points that could be extended or clarified.(1) The authors state that the model includes 21 parameters. However, they exclude a free parameter that acts as a scaling factor and is necessary to fit the experimental data (lambda). As a result, SeqDYN does not predict the spectrum from the sequence de-novo, but requires a one parameter fitting. The authors mention that this factor is necessary due to non-sequence dependent factors such as the temperature and magnetic field strength used in the experiment.Given these considerations, would it be possible to predict what this scaling factor should be based on such factors?

There are still too few data to make such a prediction.

(2) The authors mention that the Lorentzian functional form fits the data better than a Gaussian functional form, but do not present these results.

We tested the different functional forms at the early stage of the method development. The improvement of the Lorentzian over the Gaussian was slight and we simply decided on the Lorentzian and did not go back and do a systematic analysis.

(3) The authors mention that they conducted five-fold cross validation to determine if differences between amino acid parameters are statistically significant. While two pairs are mentioned in the text, there are 190 possible pairs, and it would be informative to more rigorously examine the differences between all such pairs.

We now present t-test results for other pairs in new Fig. S3.

**Reviewer #3 (Public Review):**
The manuscript by Qin and Zhou presents an approach to predict dynamical properties of an intrinsically disordered protein (IDP) from sequence alone. In particular, the authors train a simple (but useful) machine learning model to predict (rescaled) NMR R2 values from sequence. Although these R2 rates only probe some aspects of IDR dynamics and the method does not provide insight into the molecular aspects of processes that lead to perturbed dynamics, the method can be useful to guide experiments.A strength of the work is that the authors train their model on an observable that directly relates to protein dynamics. They also analyse a relatively broad set of proteins which means that one can see actual variation in accuracy across the proteins.A weakness of the work is that it is not always clear what the measured R2 rates mean. In some cases, these may include both fast and slow motions (intrinsic R2 rates and exchange contributions). This in turn means that it is actually not clear what the authors are predicting. The work would also be strengthened by making the code available (in addition to the webservice), and by making it easier to compare the accuracy on the training and testing data.

Our method predicts the sequence dependence of R2, which is dominated by slower dynamics.

**Recommendations for the authors:**

**Reviewer #2 (Recommendations For The Authors):**
(1) Should make sure to define abbreviations such as NMR and SeqDYN.

We now spell out NMR at first use. SeqDYN is the name of our method and is not an abbreviation.

(2) The authors do not mention how the curves in Figure 2A are calculated.

As we stated in the figure caption, these curves are drawn to guide the eye.

(3) May be interesting to explore how the model parameters (q) correlate with different measures of hydrophobicity (especially those derived for IDPs like Urry). This may point to a relationship between amino acid interactions and amino acid dynamics

We now present the correlation between q and a stickiness parameter refined by Tesei et al. (new ref 45) and used for predicting phase separation equilibrium (new Fig. S6).

(4) The authors demonstrate that secondary structure cannot be fully accounted for by their model. They make a correction for extended alpha-helices, but the strength of this correction seems to only be based on one sequence. Would a more rigorous secondary structure correction further improve the model and perhaps allow its transferability to ordered proteins?

We have five 4 test cases (Figs. 4E, F and 5H, I). However, we doubt that the SeqDYN method will be transferable to ordered proteins.

**Reviewer #3 (Recommendations For The Authors):**
Changes that could strengthen the manuscript substantially.(1) The authors do not really define what they mean by dynamics, but given that they train and benchmark on R2 measurements, the directly probe whatever goes into the measured R2. Using a direct measurement is a strength since it makes it clear what they are predicting. It also, however, makes it difficult to interpret. This is made clear in the text when the authors, for example write "𝑅2 is the one most affected by slower dynamics (10s of ns to 1 μs and beyond)." First, with the "and beyond" it could literally mean anything. Second, the "normal" R2 rate is limited up to motions up to the (local) "tumbling/reorganization" time (which is much faster), so any slow motions that go into R2 would be what one would normally call "exchange". The authors should thus make it clearer what exactly it is they are probing. In the end, this also depends on the origin of the experimental data, and whether the "R2" measurements are exchange-free or not. This may be a mixture, which hampers interpretations and which may also explain some of the rescaling that needs to be done.

We now remove “and beyond”, and also raise the possibility that R2 measurements based on 15N relaxation may have relatively small exchange contributions (p. 17).

(2) Related to the above, the authors might consider comparing their predictions to the relaxation experiments from Kriwacki and colleagues on a fragment of p27. In that work, the authors used dispersion experiments to probe the dynamics on different timescales. The authors would here be able to compare both to the intrinsic R2 rates (when slow motions are pulsed away) as well as the effective R2 rates (which would be the most common measurement). This would help shed light on (at least in one case) which type of R2 the prediction model captures. https://doi.org/10.1021/jacs.7b01380

We now report this comparison in new Fig. S5 and discuss its implications (p. 17-18).

(3) In some cases, disagreement between prediction and experiments is suggested to be due to differences in temperature, and hence is used as an argument for the rescaling done. Here, the authors use a factor of 2.0 to explain a difference between 278K and 298K, and a factor of 2.4 to explain the difference between 288K and 298K. It would be surprising if the temperature effect from 288K->298K is larger than from 278K->298K. Does this not suggest that the differences come as much from other sources?

Note that the scaling factors 2.0 and 2.4 were obtained on two different IDPs. It is most likely that different IDPs have different scaling factors for temperature change. As a simple model, the tumbling time for a spherical particle scales with viscosity and the particle volume; correspondingly the scaling factor for temperature change should be greater for a larger particle than for a smaller particle.

(4) The authors find (as have others before) aromatic residues to be common at/near R2 peaks. They suggest this to be indicative for Pi-Pi interactions. Could this not be other types of interactions since these residues are also "just" more hydrophobic? Also, can the authors rule out that the increased R2 rates near aromatic residues is not due to increased dynamics, but simply due to increased Rex-terms due to greater fluctuations in the chemical shifts near these residues (due to the large ring current effects).

We noted both pi-pi and cation-pi as possible interactions that raise R2. There can be other interactions involving aromatic residues, but it’s unlikely to be only hydrophobic as Arg is also in the high-*q* end. For the same reason, a ring-current based explanation would be inadequate.

(5) The authors write: "We found that, by filtering PsiPred (http://bioinf.cs.ucl.ac.uk/psipred) (35) helix propensity scores (𝑝,-.) with a very high cutoff of 0.99, the surviving helix predictions usually correspond well with residues identified by NMR as having high helix propensities." It would be good to show the evidence for this in the paper, and quantify this statement.

The cases of most interest are the ones with long predicted helices, of which there are only 3 in the training set. For Sev-NT and CBP-ID4, we already summarize the NMR data for helix identification in the first paragraph of Results; the third case is KRS-NT, which we elaborate in p. 14.

(6) When analysing the nine test proteins, it would be very useful for the reader to get a number for the average accuracy on the nine proteins and a corresponding number for the training proteins. The numbers are maybe there, but hard to find/compare. This would be important so that one can understand how well the model works on the training vs testing data.

We now present the mean RMSE comparison in p. 14.

(7) The authors write: "The 𝑞 parameters, while introduced here to characterize the propensities of amino acids to participate in local interactions, appear to correlate with the tendencies of amino acids to drive liquid-liquid phase separation." It would be good to show this data and quantify this.

We now list supporting data in p. 18 and present new Fig. S6 for further support.

(8) It is great that the authors have made a webservice available for easy access to the work. They should in my opinion also make the training code and data available, as well as the final trained model. Here it would also be useful to show the results from the use of a Gaussian that was also tested, and also state whether this model was discarded before or after examining the testing data.

We have listed the IDP characteristics and sequences in Tables S1 and S2. We’re unsure whether we can disseminate the experimental R2 data without the permission of the original authors. As for the Gaussian function, as stated above, it was abandoned at an early state, before examining the testing data.

Changes that would also be useful(1) The authors should make it clearer what they predict and what they don't. They mention transient helix formation and various contacts, but there isn't a one-to-one relationship between these structural features and R2 rates. Hence, they should make it clearer that they don't predict secondary structure and that an increased R2 rate may be indicative of many different structural/dynamical features on many different time scales.

We clearly state that we apply a helix boost after the regular SeqDYN prediction.

(2) The authors write "Instead, dynamics has emerged as a crucial link between sequence and function for IDPs" and cite their own work (reference 1) as reference for this statement. As far as I can see, that work does not study function of IDPs. Maybe the authors could cite additional work showing that the dynamics (time scales) affects function of IDPs beyond "just" structure? Otherwise, the functional consequences are not clear. Maybe the authors mean that R2 rates are indicative of (residual) structure, but that is not quite the same. Also, even in that case, there are likely more appropriate references.

Ref. 1 summarized a number of scenarios where dynamics is related to function.

(3) The authors might want to look at some of the older literature on interpreting NMR relaxation rates and consider whether some of it is worth citing.Fitting/understanding R2 profiles https://doi.org/10.1021/bi020381o
https://doi.org/10.1007/s10858-006-9026-9MD simulations and comparisons to R2 rates without ad hoc reweighting (in addition to the papers from the authors themselves). https://doi.org/10.1021/ja710366c
https://doi.org/10.1021/ja209931w

The R2 data for the two unfolded proteins are very helpful! We now present the comparison of these data to SeqDYN prediction in Fig. 6C, D. The MD papers are superseded by more recent studies (e.g., refs. 1 and 14).

There are more like these.(4) In the analysis of unfolded lysozyme, I assume that the authors are treating the methylated cysteines (which are used in the experiments) simply as cysteine. If that is the case, the authors should ideally mention this specifically.

Treatment of methylated cysteines is now stated in the Fig. 6 caption.

(5) The authors write "Pro has an excessively low ms𝑅2 [with data from only two IDPs (32, 33)], but that is due to the absence of an amide proton." It would be useful with an explanation why lacking a proton gives rise to low 15N R2 rates.

That assertion originated from ref. 32.

(6) When applying the model, the authors predict msR2 and then compare to experimental R2 by rescaling with a factor gamma. It would be good to make it clearer whether this parameter is always fitted to the experiments in all the comparisons. It would be useful to list the fitted gamma values for all the proteins (e.g. in Table S1).

We already give a summary of the scaling factors (“For 39 of the 45 IDPs, Υ values fall in the range of 0.8 to 2.0 s–1”, p. 10).

(7) p. 14 "nineth" -> "ninth"

Corrected